# Daytime SHP2 inhibitor dosing, when immune cell numbers are elevated, shrinks neurofibromas

Niousha Ahmari[1], Kwangmin Choi[1], Jianqiang Wu[1], Tilat A Rizvi[1], Mark Jackson[1], Leah J Kershner[1], Mi-Ok Kim[3], Xiyuan Zhang[4], Eva Dombi[4], Jack Shern[4], David A Hildeman[2,5], Nancy Ratner[1]

Loss of NF1 in Schwann cells leads to activation of the RAS-MAPK pathway, followed by immune cell recruitment and development of benign nerve tumors (PNFs). MEK inhibitors, which shrink most PNFs, also reduce tumor-associated myeloid cells. We tested whether SHP2 inhibition, predicted to block RAS-MAPK signaling and exert immunomodulatory effects, alters tumor volume or the immune microenvironment in PNFs, using flow cytometry and single-cell RNA sequencing. We found that both cobimetinib and daytime RMC-4550 similarly reduced tumor volume. The abundance of CD163-negative PNF-associated macrophages, derived from circulating monocytes, correlated with tumor size. Combining SHP2 inhibition with anti-PD1 altered tumor monocyte phenotype and reversed SHP2-mediated tumor shrinkage. Diurnal patterns of monocyte trafficking were disrupted in tumor-bearing mice, and SHP2 inhibition reduced tumor volume only when administered during the day, when myeloid infiltration was low. These findings suggest that SHP2 inhibitor–driven tumor shrinkage requires targeting monocyte-derived macrophages and is influenced by the timing of drug administration.

## Introduction

Benign peripheral nerve tumors known as neurofibromas arise because of biallelic inactivation of the tumor suppressor gene *Neurofibromatosis type 1* (*NF1*) in nerve Schwann cells (Serra et al, 2000; Pemov et al, 2017). A subset of neurofibromas called plexiform neurofibromas (PNFs) form in up to half of individuals with NF1. These tumors grow most rapidly in childhood and can cause severe morbidity (Fisher et al, 2022). Neurofibroma Schwann cells exhibit hyperactivation of RAS signaling, because the *NF1* gene product normally accelerates the hydrolysis of active RAS-GTP to inactive RAS-GDP (Bollag & McCormick, 1991; DeClue et al, 1992). Ras proteins are themselves central regulators of intracellular signaling that become activated (GTP-bound) in response to extracellular stimuli. Activation of receptors by extracellular ligands facilitates the recruitment of Ras guanine exchange factors (Ras-GEFs), converting inactive Ras-GDP to active Ras-GTP, thus activating downstream signaling pathways including the RAS-RAF-MEK-ERK (MAPK) pathway (Bos et al, 2007; Simanshu et al, 2017). This central pathway plays critical roles in cellular processes including cellular growth, differentiation, migration, metabolism, and immunomodulation. Indeed, MEK1/2 inhibitors elicit partial shrinkage of neurofibromas in both preclinical mouse models and clinical trials. However, some tumors are intrinsically resistant, tumors rarely shrink beyond 20%, and tumors regrow upon treatment cessation (Jessen et al, 2013; Dombi et al, 2016; Gross et al, 2020; Weiss et al, 2021), highlighting an unmet need for more effective or combinatorial strategies.

The neurofibroma microenvironment is characterized by a unique supportive stroma (Jiang et al, 2023) and by a high density of macrophages (Prada et al, 2013; Ribeiro et al, 2013; Fletcher et al, 2019). PNFs arise in peripheral nerves, and 30% of the total cells are myeloid cells that can adopt various phenotypes depending on local signals (Prada et al, 2013; Ribeiro et al, 2013; Farschtschi et al, 2016; Haworth et al, 2017; Kershner et al, 2022). At steady state, peripheral nerve macrophages are replenished from blood monocytes and differentiate into functionally distinct macrophage subsets (Lund et al, 2024). Early and late in tumor formation, macrophages exert different effects on neurofibroma growth. Thus, blocking colony-stimulating factor 1 receptor (CSF1R) reduces monocytes/macrophages early in tumorigenesis and restrains PNF growth, yet the same approach later in tumor development supports tumor expansion (Prada et al, 2013). In benign neurofibromas, T cells account for only up to 2% of cells. Remarkably, however, CD8 T cells paradoxically enhance rather than inhibit tumorigenesis, and depletion of PNF T cells results in a reduction of PNF macrophages (Fletcher et al, 2019; Pundavela et al, 2024). In other settings, T cells also modulate macrophage function (Endig et al, 2016; Roberts et al, 2007). In some mouse models of NF1, a heterozygous environment can increase tumor size (Yang et al, 2008; Brosseau et al, 2018).

[1]Divisions of Experimental Hematology and Cancer Biology, Cancer and Blood Diseases Institute, Cincinnati Children's Hospital Medical Center, Cincinnati, OH, USA [2]Division of Immunobiology, Cincinnati Children's Hospital Medical Center, Cincinnati, OH, USA [3]Department of Epidemiology and Biostatistics, UCSF, San Francisco, CA, USA [4]Pediatric Oncology Branch, National Cancer Institute, Bethesda, MD, USA [5]Department of Pediatrics, University of Cincinnati College of Medicine, Cincinnati, OH, USA

Correspondence: nancy.ratner@cchmc.org

However, in the DhhCre;Nf1fl/fl model, immune cells are WT, and tumor forms in all mice (Wu et al, 2008). How T cells and macrophages each contribute to neurofibroma growth and how to modulate each cell type for therapeutic benefit remain unclear.

Upstream of MEK, Src homology 2 containing phosphatase (SHP2) acts by dephosphorylating the RAS exchange factor SOS1, thus activating the RAS pathway (Nichols et al, 2018; Bos et al, 2007; Simanshu et al, 2017; reviewed in Sodir et al [2023]). SHP2 inhibitors that were initially developed to target oncogenic signaling in tumors are now known to confer additional immunomodulatory effects, influencing macrophage subsets, depleting pro-tumorigenic populations, and enhancing tumor sensitivity to additional immunotherapies (Lorenz, 2009; Xiao et al, 2018; Christofides et al, 2023). Allosteric SHP2 inhibitors, many now in clinical testing, also suppress RAS-MEK-ERK signaling in diverse malignancies, including several with *NF1* loss (Fedele et al, 2018; Nichols et al, 2018; Drilon et al, 2023; Sait et al, 2025). In NF1 mutant malignant peripheral nerve sheath tumor (MPNST) models, SHP2 inhibition transiently reduced tumor size, and combining it with further MAPK pathway inhibition increased durability (Wang et al, 2020, 2023; Wei et al, 2023). These results suggest that targeting the upstream RAS regulator SHP2 can improve therapeutic outcomes not only by dampening oncogenic RAS-MAPK signaling, but also by modifying the tumor immune landscape. In an immunogenic syngeneic preclinical cancer model, SHP2 inhibition drove direct and selective depletion of pro-tumorigenic macrophages via attenuation of CSF1 receptor signaling and increased antitumorigenic macrophages via a mechanism independent of CD8⁺ T cells or IFNγ (Quintana et al, 2020). Importantly, in a preclinical cancer model, ablating SHP2 in myeloid cells reduced tumor growth, whereas lymphocyte-specific deletion did not (Christofides et al, 2023). However, the immunomodulatory capacity of RAS-MAPK pathway inhibitors varies with the agent used and the immune microenvironment in which the tumor arises (Fedele et al, 2018; Quintana et al, 2020; Christofides et al, 2023), emphasizing the need to study specific tumor types to identify relevant modes of therapeutic immunomodulation.

We hypothesized that inhibiting SHP2 would shrink neurofibromas and normalize aberrant immune cell activity in neurofibromas. We find that although both MEK and SHP2 inhibitors shrink PNF tumors and dramatically reduce myeloid cells in tumors, SHP2 inhibition more profoundly alters TAMs and increases tumor T-cell infiltration. Importantly, we identify altered local and systemic diurnal immune perturbations in animals with PNF and find that the immunomodulatory effects of SHP2 inhibition and tumor shrinkage depend on the timing of drug delivery and correlate with immune cell abundance within tumors. Our results demonstrate the therapeutic potential of SHP2 inhibition as a single agent in neurofibromas and the immunomodulatory potential of these small molecules.

# Results

## SHP2 inhibition, like MEK inhibition, reduces murine PNF tumor volume

We tested whether the SHP2 inhibitor RMC-4550 is active, alone or in combination with MEK inhibition, at shrinking tumors in a robust murine model of PNF, DhhCre;Nf1^fl/fl^ (Wu et al, 2008; Jessen et al, 2013). In this model, *Nf1* loss in the SC lineage at embryonic day 12.5 (before terminal glial differentiation) results in PNF in all mice by 4 mo of age; these tumors do not transform to malignancy. We treated DhhCre;Nf1fl/fl mice with a MEK inhibitor (5 mg/kg GDC-0973/cobimetinib; q.d., p.o., MEKi), a SHP2 inhibitor (10 or 30 mg/kg RMC-4550; q.d., p.o., SHP2i), or the combination, each on a 5-d-on, 2-d-off schedule. Tumors in vehicle-treated mice grew over the 60-d treatment period (Fig 1A). Single-agent treatment with MEK inhibitor, and each of the two doses of SHP2 inhibitor, significantly reduced tumor volume as assessed by random-effects analysis ($P < 0.0001$). This mixed model analysis allows analysis of volume change, as it accounts for the known heterogeneity in neurofibroma growth in the *DhhCre;Nf1fl/fl* mouse model; heterogeneity in change in tumor size is observed both across a group and in longitudinal measurements for each mouse (Wu et al, 2012). Interestingly, combined inhibition of MEK and SHP2 was not better than single inhibitor therapy (MEK inhibitor alone [$P = 0.1959$] or SHP2 inhibitor alone [$P > 0.9999$]). We then assessed the percentage of tumors that shrank during the treatment period (from month 7 to month 9). Tumors shrank in most treated mice (13/14 in the combo-treated mice, 13/18 in MEK inhibitor–treated mice, and 18/21 in SHP2 inhibitor–treated mice). The percentage of mice with tumor shrinkage was significantly higher in all treatment groups compared with the vehicle controls (*P*-values for all three comparisons versus vehicle, $P < 0.0001$) but was similar among the non–vehicle-treated mouse groups (all *P*-values for comparisons among the treated groups [single or combo], $P > 0.1$). Plasma levels of RMC-4550 (Fig 1D, left panel) were similar in the combination treatment group, 2 and 8 h after the final dose of the compounds, as compared to RMC-4550 treatment alone. Plasma levels of cobimetinib increased in the combination as compared to cobimetinib alone (Fig 1B, right panel), but the combination, even with slightly elevated plasma levels of cobimetinib, did not provide additional benefit. Body weight, monitored as a general indicator of tolerability (defined as >10% change in body weight), was not significantly changed in any treatment group (Fig 1C). We then compared SHP2 and combination effects on the MAPK pathway. As a pharmacodynamic readout, we analyzed pERK in tumor lysates, 2 h after a final drug dose, using Western blotting. The ratio of total to pERK was similarly reduced in SHP2 inhibitor– and combination-treated mouse tumors, in contrast to lung tissue in which the combination showed more reduction than SHP2 inhibitor alone (Fig 1D). Immunostaining of paraffin sections verified the reduced pERK in inhibitor-treated tumors and showed a significant reduction in the percentage of Ki67+ tumor cells in all treatment groups when compared to vehicle controls; there was no significant difference with the combination therapy (Fig 1E and F). Thus, MEK inhibition and SHP2 inhibition are active as single agents at reducing tumor volume in the DhhCre;Nf1fl/fl mouse model of PNF.

MEK inhibition is not durable in mice or in individuals with PNF (Jousma et al, 2015; Dombi et al, 2016; Gross et al, 2020). We tested the durability of single agents, treating mice for 1 mo with MEK (1.5 mg/kg PD0325901) or SHP2 inhibitor (10 mg/kg), and then maintaining mice under off treatment for an additional 1 month, monitoring tumor volume. Tumors grew back in both treatment

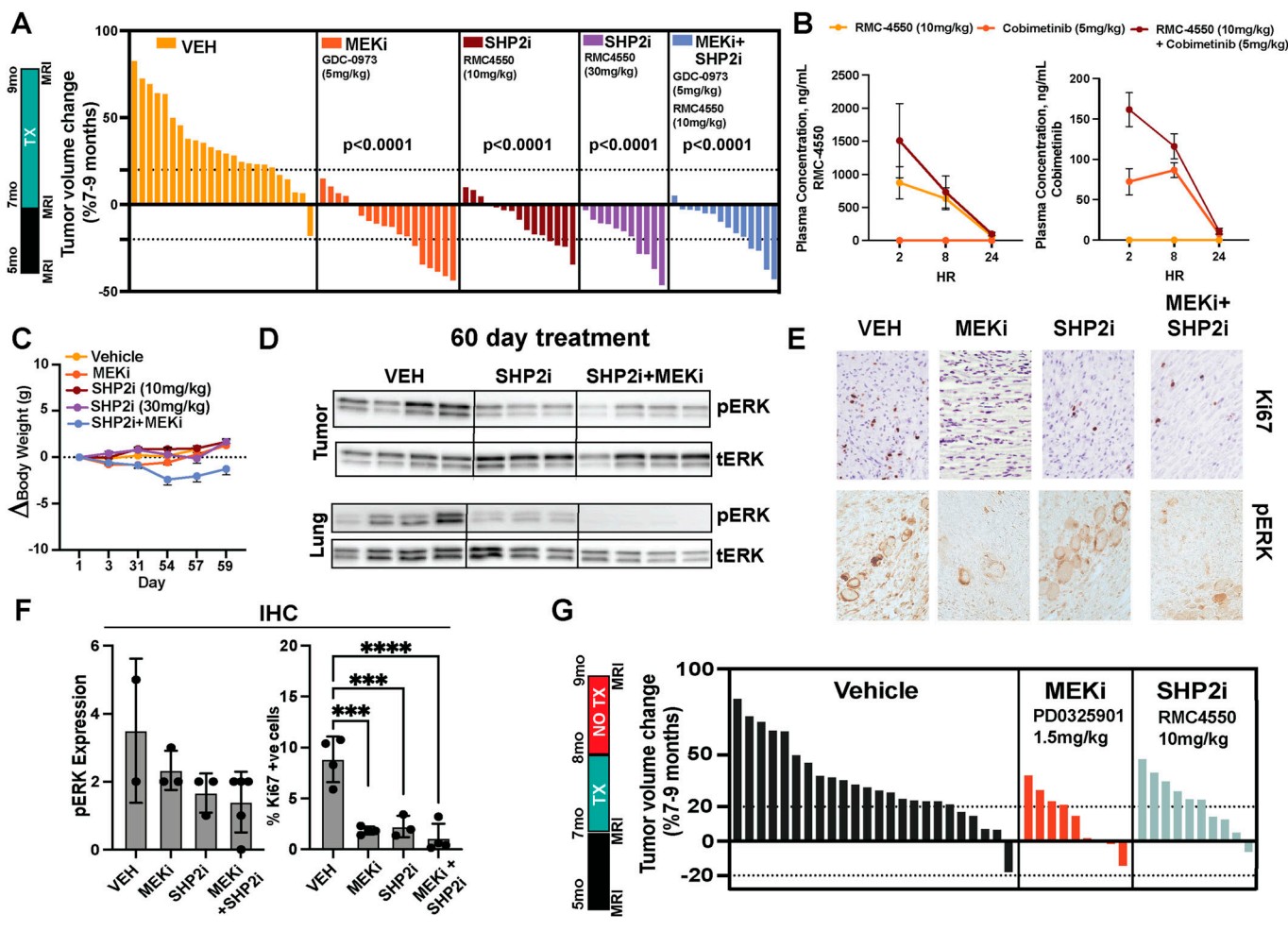

**Figure 1. SHP2 inhibition shows single-agent activity in neurofibroma, which is not increased by combination with MEK inhibition and is not durable.**
**(A)** Waterfall plot illustrating percent change in tumor volume during the 60-d dosing period based on consecutive MRI scans, demonstrating similar tumor shrinkage in treatment groups. Each bar represents the percent change in tumor volume for an individual mouse. Dotted lines indicate ±20% change. **(B)** Plasma concentration (nM) after administration of cobimetinib (5 mg/kg) and RMC-4550 (10 mg/kg) measured over 24 h after a final dose. **(C)** Percent change in body weight by the treatment group. **(D)** Western blot reveals showing phospho-ERK in tumors from mice receiving vehicle or combination treatment (10 mg/kg RMC-4550 and 5 mg/kg cobimetinib) compared with single-agent treatment (10 mg/kg RMC-4550). SHP2i in combination with MEKi shows additive activity in lung, but not in tumors. **(E)** Representative images of Ki67 immunohistochemistry staining in tumor sections from vehicle treated 60 d with either Vehicle, RMC-4550 (10 mg/kg), or RMC-4550 (10 mg/kg) and cobimetinib (5 mg/kg). **(F)** Quantification of phospho-ERK staining intensity (left) and percentage of Ki67-positive cells (right) in tumors from mice after 60 d of treatment. **(G)** Waterfall plot illustrating the percent change in tumor volume during a 60-d period (~30-d treatment followed by 30-d off drug), imaged using consecutive MRI scans. Each bar represents data from a single mouse. Tumors grew in all groups.
Source data are available for this figure.

groups. Tumors in each treatment group remained smaller than vehicle-treated tumors, suggesting regrowth without rebound, but not a durable response (Fig 1G).

### Single-cell RNA-seq highlights reduction in tumor immune cells and immune cell transcriptional changes caused by MEK and SHP2 inhibition

To gain insight into transcriptional changes in tumor cell populations, we characterized the effects of MEKi and SHP2i using single-cell RNA-seq (10x Genomics, 3′ v3.1 chemistry). We treated tumor-bearing mice for 30 d with vehicle (n = 5) or each single agent (MEKi (PD0325901; 1.5 mg/kg; p.o. n = 4), SHP2i (n = 3)); 3,386–9,627 cells/tumor were used for scRNA-seq. Sequencing depth per

capture was ~400 million reads. Using Seurat, we identified 26 stable cell clusters at a resolution of 0.6 (Fig 2A). Immune cells were significantly reduced as a percentage of total live cells by treatment with MEK inhibitor or SHP2 inhibitor, from 41% of total tumor cells in vehicle-treated tumors to 6.3% in MEKi-treated and 5.3% in SHP2i-treated tumors (Fig 2B). In contrast, the relative cluster sizes of types of Schwann cells (Fig 2C) and of types of immune cells were not dramatically changed (Fig 2D). The pathways affected by inhibitor responses versus vehicle were highlighted by a gene set enrichment analysis using Enrichr (Fig 2E). Table S1 shows pathway genes altered in each cluster. Importantly, no gene showed expression changes uniquely in one of the two treatment groups; rather, the significance of changes differed. The most changes were in C21, containing cells with characteristic

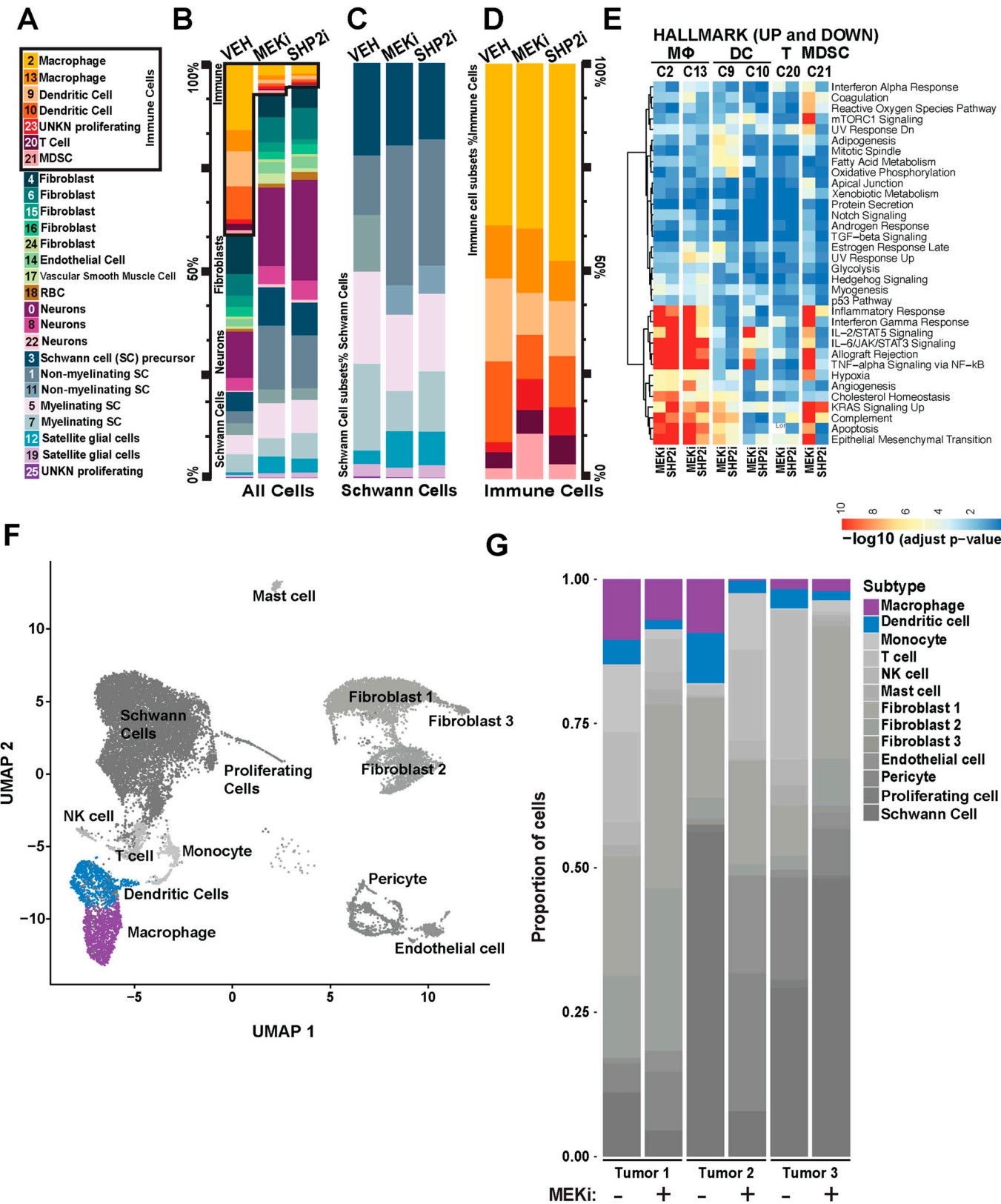

**Figure 2. SHP2 and MEK inhibition each shrink the proportion and alter the transcriptomes of neurofibroma immune cells.**
**(A)** Cluster identification from single-cell analysis comparing tumor-bearing mice treated with vehicle (VEH), MEK inhibitor (MEKi, PD0325901), or SHP2 inhibitor (SHP2i, RMC-4550). **(B)** Proportion plots show single-cell RNA-seq data, presented as the percentage of each cell type, highlighting reduced proportion of plexiform neurofibroma immune cell clusters in drug-treated mice. Immune cell populations are boxed in black. **(C)** Proportion plots show single-cell RNA-seq data, presented as

markers of changes of myeloid-derived suppressor cells (MDSCs), which can be derived from diverse monocytic cells with dendritic cells or neutrophil-like phenotypes (Hegde et al, 2021; Veglia et al, 2021). Other changes were in C10, a dendritic cell cluster, and C13, a macrophage cluster.

A recent report used bulk RNA sequencing to define the changes that occur in human PN while under treatment with the MEKi selumetinib (Gross et al, 2025). Consistent with our mouse single-cell data, the results of this study demonstrated significant changes to the myeloid compartment of the TME while under the therapeutic pressure of MEKi. To enable more direct comparisons between the preclinical mouse model and the human tumors, we performed single-cell RNA sequencing of three human PN taken from biopsy specimens of two adult NF1 patients before and after the completion of their first cycle of MEK inhibitor (six total specimens; see Fig S1A). A total of 21,022 cells (10,566 cells from tumors before the treatment and 10,456 cells from tumors being treated) passed quality control and were included in the downstream analysis. The analysis identified 13 cellular clusters including tumorigenic Schwann cells, and immune and stromal compartments at resolution 0.1 (Figs 2F and G and S1B). Consistent with the results from the mouse studies, we observed that the percentage of detected Schwann cells was not dramatically changed in the treated tumors. In contrast, there were notable contractions in the macrophage and dendritic cell populations in the on-treatment tumors. Given the concordance between the human and mouse neurofibroma response to MEKi, we decided to perform deeper profiling and analyze effects of SHP2 inhibition in the more tractable murine system.

More detailed analysis of immune cell populations is shown in Fig S2, with defining markers of transcription in myeloid immune cells in Fig S2A. Cluster-by-cluster analysis revealed inhibitor-driven differences in transcription in macrophage C13, dendritic cells C9 and C10, T-cell cluster C20, and MDSC C21, which in MEKi-treated cells showed two patterns. Some cells retained expression patterns of vehicle-treated cells, whereas others showed a novel signature (Fig S2B). In contrast, after SHP2i treatment, all cells showed the de novo signature (Fig S2B). We then compared WT and tumor macrophage clusters (Fig S2C), and tumor macrophages after vehicle versus drug treatment (Fig S2D and E). *Cd163* expression was reduced in tumor macrophages and showed relative increases after drug treatment (Fig S2C and D). Changes in transcription of markers of activation were also caused by drug treatment (Fig S2E). Schwann cell clusters showed minimal changes in RNA expression and similar changes after MEK and SHP inhibition (Fig S2F). Thus, MEK and SHP2 inhibition each have robust transcriptional effects on tumor immune cells, and SHP2 inhibition results in more homogeneous, and smaller, immune cell populations.

To independently validate and quantify these findings, we next used high-dimensional flow cytometry (Fig 3) to analyze immune populations based on surface marker expression. Unlike scRNA-seq, which captures relative proportions of immune cells within the total tumor cell population, flow cytometry enables direct measurement of both absolute and relative changes in defined immune subsets. Fig 4 builds upon these flow cytometry data by resolving macrophage subpopulations and assessing their phenotypes using unsupervised clustering and morphometric analysis. Together, these complementary methods reinforce the conclusion that SHP2 inhibition broadly reduces immune cell abundance—particularly CD163⁻ macrophages—while revealing changes in composition and activation state among the residual immune populations. To independently validate and quantify these findings, we next used high-dimensional flow cytometry (Fig 3) to analyze immune populations based on surface marker expression. Unlike scRNA-seq, which captures relative proportions of immune cells within the total tumor cell population, flow cytometry enables direct measurement of both absolute and relative changes in defined immune subsets. Fig 4 builds upon these flow cytometry data by resolving macrophage subpopulations and assessing their phenotypes using unsupervised clustering and morphometric analysis. Together, these complementary methods reinforce the conclusion that SHP2 inhibition broadly reduces immune cell abundance—particularly CD163⁻ macrophages—while revealing changes in composition and activation state among the residual immune populations.

## Multidimensional flow cytometry reveals different effects on tumor immune cells caused by MEK and SHP2 inhibition

To further define how MEK inhibition and SHP2 inhibition affect immune cells, we used flow cytometry. We treated tumor-bearing mice for 30 d with vehicle (n = 11) or each single agent (MEKi [PD0325901; 1.5 mg/kg; p.o. n = 5], SHP2i [n = 710 mg/kg; n = 4 30 mg/kg]), examining tumors after 30 d of treatment. Multi-parametric spectral flow cytometry confirmed significant reductions in immune cells after each inhibitor treatment and emphasized major effects on macrophage number. Cell-type designations used an antibody panel and gating strategy that distinguishes immune and nonimmune cells and distinguishes differentiated macrophages from dendritic cells (DCs), based on a framework described for tissues with resident macrophages (Lavin et al, 2017; Fig S3A). Validating this analysis, cell proportions in WT nerve/DRG were equivalent to those described in a recent study of nerve and DRG (Lund et al, 2024). As in single-cell analysis, we identified Fig 3A shows sunburst plots that visualize hierarchy based on gating strategy, highlighting altered cell proportions in the WT nerve/DRG versus tumors, and after drug exposure.

the percentage of cells within each Schwann cell cluster, as a percentage of all glial cells. **(D)** Proportion plots show single-cell RNA-seq data, presented as the percentage of cells within each immune cell cluster, as a percentage of all immune cells. **(E)** Gene set enrichment ($-\log_{10}$(adjusted *P*-value)) for Schwann cell clusters for MEKi (PD0325901) or SHP2i (RMC-4550), each versus vehicle. **(F)** UMAP plot showing clustering of cell types from combined scRNA-seq data across three human PN tumors, with or without selumetinib treatment. Major immune and stromal populations were annotated based on canonical marker expression. **(G)** Stacked bar plots showing the relative proportions of each annotated cell type in individual tumors pre- and post-treatment with selumetinib. Selumetinib treatment is associated with a marked decrease in multiple immune populations, including macrophages, dendritic cells, T cells, and NK cells.

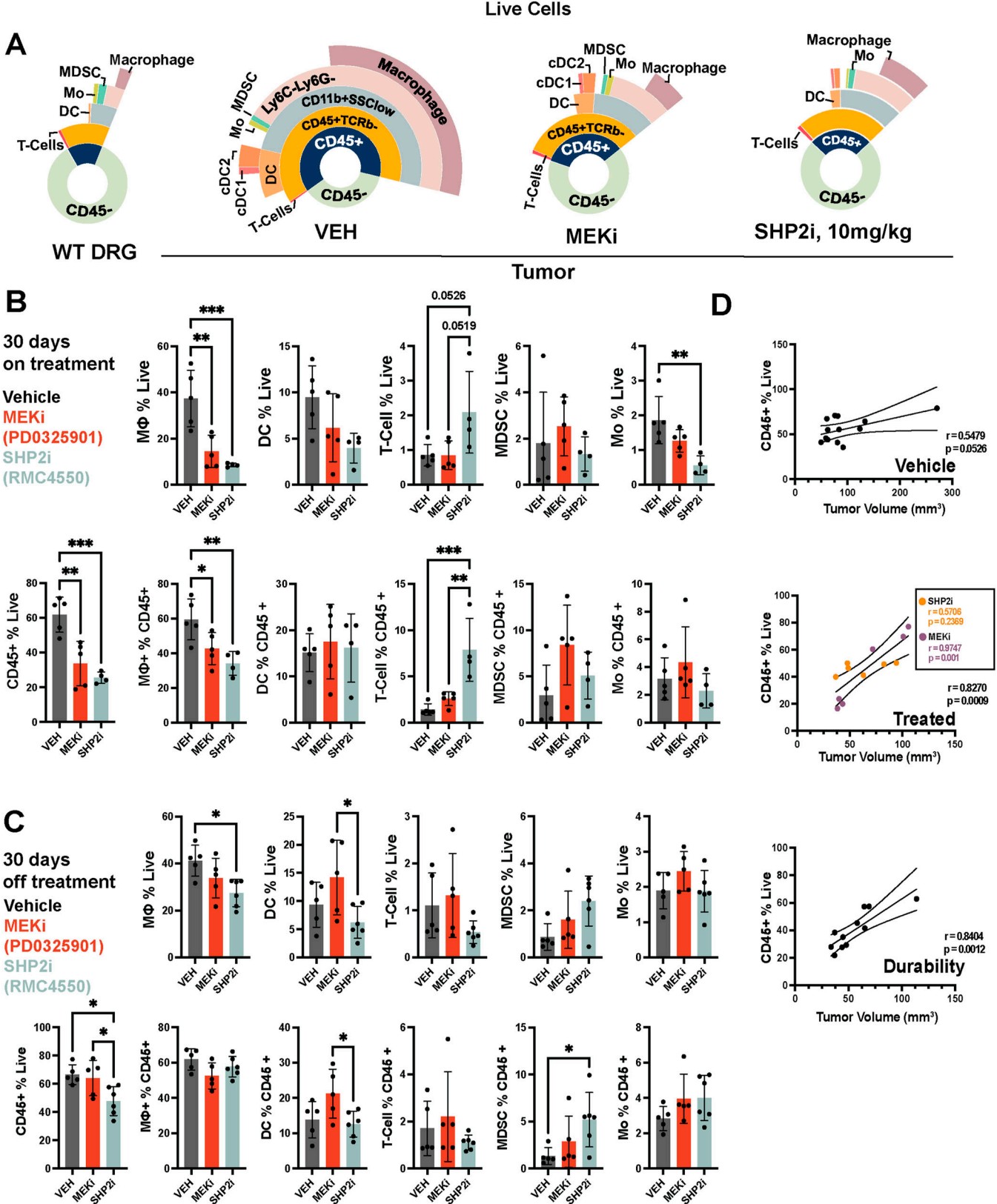

**Figure 3. Immune cell landscape in normal nerve/DRG and plexiform neurofibroma analyzed by high-dimensional flow cytometry reveals sustained effects of SHP2 inhibition on plexiform neurofibroma.**

**(A)** Sunburst plots show cell proportions in control (dorsal root ganglia/nerve from non–tumor-bearing mice (WT DRG)) versus tumor from vehicle-treated mice (VEH) treated with MEK inhibitor (MEKi, PD0325901) or SHP2 inhibitor (SHP2i, RMC-4550), shown as the percentage of live cells in each condition. Data shown are from 3 to

CD14 and CD172a distinguished nerve/PNF macrophages from CD11c+;MHCII+;CD11b-;CD172a- cDC1 and CD11c+;MHCII+;CD11b+; CD172a+ cDC2 dendritic cells (Fig S4F and H); macrophages were resolved into a CD163+ population and a CD163⁻ macrophage population that specifically and dramatically expanded in PNF versus normal nerve/DRG (Fig S4A–E; highlighted in Fig S4E). MDSC-like cells (CD11b+;Ly6C+;Ly6G+) were separated from monocytes (CD11b+;Ly6C+;Ly6G-) (Fig S4F and G). The reduction in tumor immune cells upon MEK inhibition or SHP2 inhibition at 10 mg/kg was accounted for largely by a twofold reduction in tissue macrophages, a trend toward a reduction in DCs, and a reduction in monocytes (Fig 3B). Increases in T cells were uniquely observed after SHP2 inhibitor treatment. As a percentage of total immune cells, these reductions in macrophages and increases in T cells remained significant. In sum, flow cytometry characterized immune populations in neurofibromas and verified that the reduced immune cells in neurofibromas after drug treatment are largely tumor macrophages.

We next compared immune cell population in tumors from mice treated for 30 d, followed by 30-d off treatment, when tumors regrew. After MEK inhibitor withdrawal, numbers of tumor immune cells largely returned to levels present in vehicle-treated tumors, except that prior MEK inhibition resulted in a sustained (30-d) increase in dendritic cells. In contrast, SHP2 inhibition caused a sustained decrease in tumor immune cells (Fig 3C). We wondered whether immune cells correlate with tumor volume. The percentage of tumor CD45+ immune cells correlated with tumor volume in untreated mice (Fig 3D, top; r = 0.5479). In MEK or SHP2 inhibitor–treated mice, the correlation improved (Fig 3D, middle; r = 0.827). Even after a month off therapy (Fig 3D, bottom), the correlation remained (r = 0.804).

## SHP2 inhibition selectively targets CD163⁻ macrophages

Flow cytometry emphasized that CD163⁻ macrophages preferentially increase in tumors, correlating with reduced *Cd163* expression in macrophages. CD163⁻ macrophages are derived from circulating monocytes that enter nerve (Ydens et al, 2020; Lund et al, 2024). To facilitate the confirmation of cell cluster identities, we carried out unsupervised UMAP clustering of CD45+ cells, followed by FlowSOM marker analysis. We then used Cluster Explorer and Marker Enrichment Modeling. This analysis verified that CD163⁻ macrophages were increased in the tumor versus normal DRG/nerve. CD163⁻ macrophage abundance decreased significantly (P < 0.0001) in response to MEK or SHP2 inhibition as a proportion of live cells (Fig 4B and C), and as a proportion of CD45+ immune cells (Fig 4D and E). Fig S5A–C shows the T-REX validation of the clustering approach used in the main figure. In contrast, tissue-

resident CD163+ macrophages were only slightly reduced by SHP2 inhibition (P = 0.05). To test whether inhibitor treatment kills macrophages, we removed cells from tumors and cultured them for up to 36 h in drug, then analyzed macrophages by flow cytometry. In this acute test, MEK inhibition but not SHP2 inhibition increased cell death by 36 h and affected each macrophage subset (Fig S5D and E). SHP2 inhibition might reduce macrophage abundance by altering cell trafficking and/or other mechanisms.

To ensure that drug effects on tumor macrophages are present in intact tissue, we analyzed macrophage morphology in tumor paraffin sections. Changes in macrophage shape correlate with altered cytokine production; smaller, rounder macrophages are associated with longer patient survival (McWhorter et al, 2013; Donadon, et al, 2020). Morphometric analysis of Iba1-positive macrophages in tumors from mice after 60 d of treatment revealed compound-induced changes in cell complexity. Thus, K-means clustering defined subpopulations (k = 4) based on cell area, perimeter, circularity, and roundness. Most Iba1+ cells in tumors treated with SHP2 inhibitor were found in Cluster 2, containing the smallest, roundest cells (Fig 4F and G). MEK inhibitor had a lesser effect. The Sholl analysis on the same dataset validated a reduction in macrophage process number and length upon treatment (Fig S5F–H). Thus, macrophage phenotype is differentially affected by MEK versus SHP2 inhibition in unperturbed tissue. To determine whether some of the effects of SHP2 inhibition on macrophage morphology are cell-autonomous (e.g., might occur before interaction with tumor cells), we treated BMDM (Iba1; CD172a+) for 8 h with vehicle or SHP2 inhibitor; SHP2 inhibition increased the small, round morphology (Cluster 4) versus vehicle (Fig S5I–K). Thus, SHP2i treatment targets CD163⁻ tumor macrophages, and robustly alters macrophage shape, a marker of macrophage activation.

## Monocytes from tumor-bearing mice show abnormal phenotypes

We correlated tumor volume assessed by MRI to macrophage populations assessed by flow cytometry (Fig 5A and B). A positive correlation (r = 0.5716) was observed between CD163⁻ tumor macrophages as a percentage of live cells in tumors and tumor volume; CD163+ macrophages were anticorrelated with tumor volume, but only as a percentage of CD45+ cells, suggesting modulation of resident macrophage production. To determine whether other immune populations correlate with tumor volume, we conducted principal component analysis (PCA using the R function prcomp). PC1 showed a strong correlation to tumor volume and changes in tumor volume over time (Fig 5C). Twenty immune subsets identified by flow cytometry were correlated with tumor volume. The top cell types comprising PC1 were myeloid

---

5 mice/treatment group. Cells are categorized into myeloid-derived suppressor cells (MDSC), macrophages, dendritic cells (DC), and T cells based on the expression of CD45 and markers per group (Figs S3 and S4). **(B)** Bar graphs display the percentage of CD45+ immune cells, and as a percentage of CD45+ cells, percentages of macrophages (as defined in Figs S3 and S4), TCRb+ T cells, CD11c+ dendritic cells (DC), MDSC, and monocytes (Mo) after 30 d of treatment with VEH, MEKi (PD0325901), or SHP2i (RMC-4550). **(B, C)** Populations of cells shown in (B), from tumor-bearing mice 30 d after a 30-d treatment period. Unless otherwise noted, each datapoint represents an individual biological replicate, and statistical significance was identified using a one-way ANOVA with Tukey's post hoc test; average ± SD is indicated with asterisks (*P < 0.05, **P < 0.01, ***P < 0.001). **(D)** Correlation between the percentage of CD45+ immune cells and tumor volume in vehicle-treated versus treated tumors. Top: vehicle-treated tumors. Middle: tumors treated with either MEK inhibitor (selumetinib) (purple) or SHP2 inhibitor (RMC-4550) (orange). Bottom: tumors in the durability group, treated with RMC-4550 for 30 d followed by a 30-d treatment-free period. Correlation coefficients (r) and P-values are shown for each group.

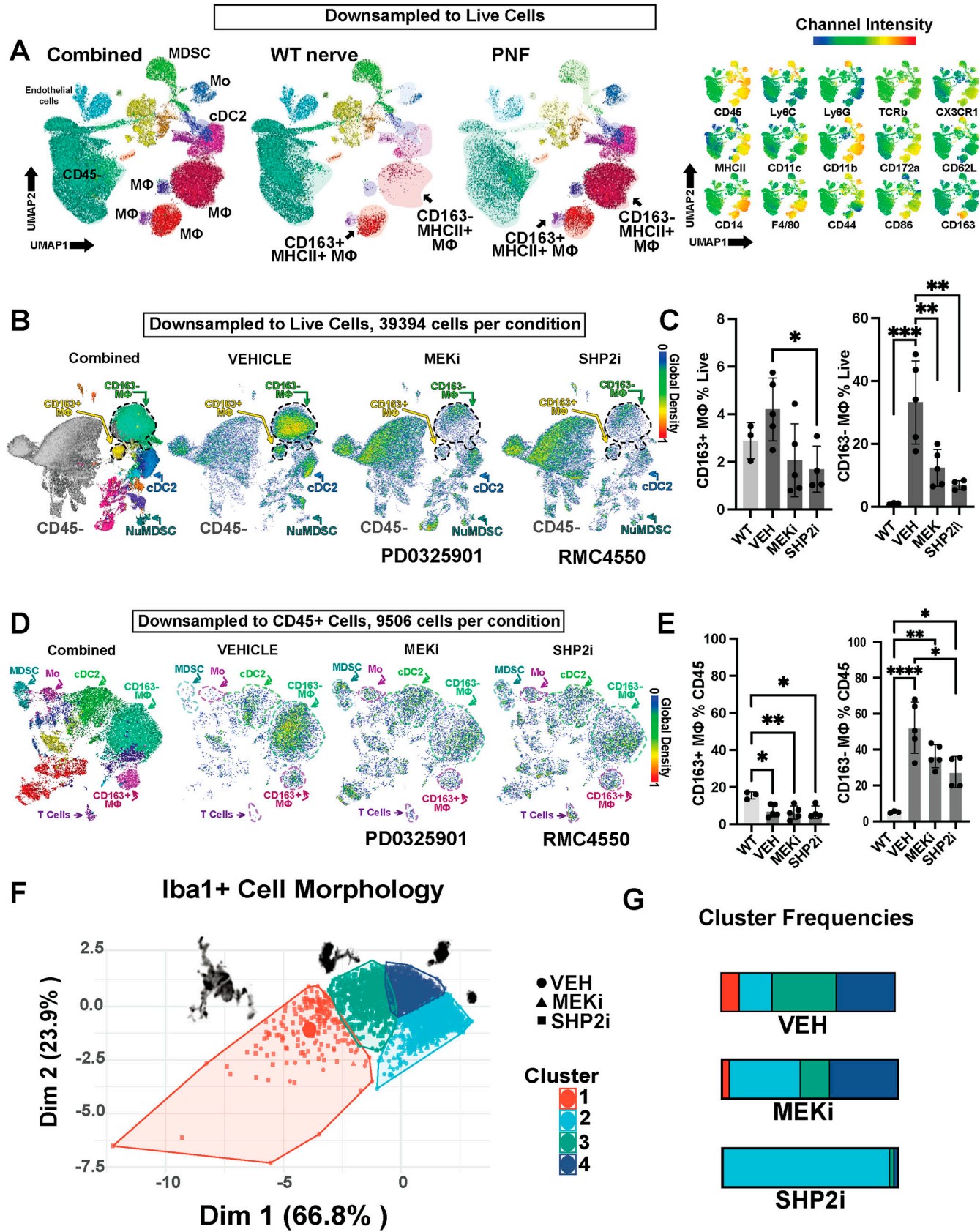

cells, CD163⁻;MHCII+ macrophages (r = 0.5716, *P* = 0.0003), and monocytes (Fig 5D). These results led us to further characterize tumor monocytes.

As noted above, at steady state, CD163⁻ peripheral nerve macrophages are rapidly replenished from blood monocytes (Lund et al, 2024). The phenotype of tumor monocytes differed from those of normal nerve/DRG. Thus, Lyc6$^{hi}$;CD62L+ monocytes in tumors decreased versus normal nerve/DRG, whereas Ly6C$^{low}$; CD62L⁻ monocytes increased (Fig 5E and F). In the DhhCre;Nf1fl/fl model, immune cells including monocytes are WT at the *Nf1* locus. We therefore wondered whether monocytes in tumor-bearing mice show altered phenotypes. Monocytes isolated from the bone marrow of PNF-bearing mice exhibited an exaggerated ex vivo response to lipopolysaccharide (LPS) challenge (Fig 5G). LPS is a component of Gram-negative bacteria and a potent inducer of innate immune responses via Toll-like receptor 4 (TLR4) signaling, and increased response to LPS is often observed in chronic inflammatory models, including cancer (Naler et al, 2022; Zhu et al, 2023). Ly6C$^{low}$ monocyte survival and activation are regulated by the orphan nuclear receptor Nur77 (Hanna et al, 2011). Treatment with an SHP2 inhibitor, but not a MEK inhibitor, significantly reduced LPS-induced expression of Nur77 (Fig 5H). Thus, monocyte and macrophage abundance positively correlate with PNF volume. Monocytes from tumor-bearing mice are fundamentally different from those in non–tumor-bearing mice, and SHP2 inhibition exerts cell-autonomous effects on these altered monocytes from tumor-bearing mice. MEK inhibition, in contrast, did not exert cell-autonomous effects on these altered monocytes, and did not affect monocyte abundance in tumors (Fig 3B).

### Anti-PD1 reverses the therapeutic effects of SHP2 inhibition

Treatment with SHP2 inhibitors has been reported to sensitize tumors to immunotherapy, leading to potentially improved therapeutic outcomes (Christofides et al, 2023). scRNA-seq analysis data showed decreases in the expression of immune checkpoint marker gene expression by drug-treated T cells, exemplified by reduction in *Clta4* and *Pdcd1* (PD1) transcripts, and PDL1 expression in macrophages was reduced after treatment with each inhibitor (Figs 6A and S6B). However, PD1 expression on T cells was maintained after SHP2 inhibitor treatment, and on monocytes and macrophages (Fig S6A). Myeloid-specific PD-1 ablation changes the fate of myeloid cells, alters T-cell phenotype, and modulates tumor growth (Strauss et al, 2020; Christofides et al, 2023). In neurofibromas, CD8 T cells enable neurofibroma growth (Pundavela et al,

2024). The expression of other T-cell activation markers in SHP2 inhibitor–treated tumors was reduced, correlating with tumor response to therapy (Fig 1). Our findings that SHP2 inhibition alters monocyte and macrophage phenotypes (Figs 2, 3, and 4) together with these results suggested that SHP2 inhibitor treatment plus anti-PD1 might further modulate T-cell and/or myeloid phenotypes and alter drug response. To test this idea, we administered RMC-4550, anti-PD1, or the combination to tumor-bearing DhhCre;Nf1fl/fl mice for 60 d and measured tumor volume using consecutive volumetric MRI scans. Mice treated with vehicle or with weekly anti-PD1, a checkpoint inhibitor used to activate tumor T cells (Topalian et al, 2016; Molgora et al, 2020), showed growing tumors, whereas SHP2 inhibitor shrank most tumors. Combination therapy prevented effects of SHP2 inhibition on tumor shrinkage (Fig 6B). In the context of SHP2 inhibition, the addition of anti-PD1 did not affect the abundance of CD4 or CD8 Teff cells (Fig 6C). Rather, although overall monocyte and macrophage numbers were unchanged, combining anti-PD1 together with SHP2 inhibitor changed myeloid cell phenotypes, reducing the frequency of Ly6C$^{low}$;CD62L- monocytes and increasing the frequency of Ly6C$^{hi}$;CD62L+ monocytes in tumors (Figs 6D and S6C) and decreasing activated (CD86⁺;CD163⁻) macrophages (Figs 6E and S6D). Thus, shifts in myeloid phenotypes correlated with resistance to therapy. We therefore posited that other factors that influence myeloid cell phenotype would similarly correlate to therapy resistance. Monocyte and macrophage phenotypes, and immune cell production, release, and infiltration into tissues, show a diurnal pattern, with pro-inflammatory immune cell production and release from the bone marrow increasing when animals are awake (reviewed in Mok et al [2024]).

### Day versus night administration of SHP2 inhibitor alters both immune responses and tumor shrinkage

We therefore tested whether neurofibroma immune cell abundance and phenotype differed in tumors depending on the time of day they were collected. We found that subsets of dendritic cells and types of myeloid-derived suppressor cells within tumors, which can differentiate from monocytes (Hegde et al, 2021), were significantly affected by time of day of collection (Figs 7A and S7A and B). We also detected a shift in the phenotype of CD45⁻ (nonimmune) cells, largely Schwann cells and fibroblasts. Based on this result, we hypothesized that we would alter response to SHP2 inhibition if we administered SHP2i when pro-inflammatory immune cells are reduced (during the day in these nocturnal animals) versus when immune cells are relatively elevated (at

**Figure 4. SHP2 inhibition alters activation of tumor macrophages, which are predominately CD163-negative.**
**(A)** UMAP analysis of immune cells from WT dorsal root ganglia/nerve (WT nerve) and plexiform neurofibromas (PNFs), after live-cell downsampling. Clusters include macrophages (MΦ), conventional dendritic cells (cDC2), monocytes (Mo), and myeloid-derived suppressor cells (MDSCs). Representative channel intensity plots demonstrate marker expression across these populations. **(B)** UMAP visualization of immune cell subsets in PNFs from vehicle(Vehicle)-, MEK inhibitor (MEKi, PD0325901)–, and SHP2 inhibitor (SHP2i, RMC-4550)–treated mice, downsampled to an equal number of live cells (39,394 per condition). **(C)** Bar graphs showing the percentage of CD163⁺ and CD163⁻ macrophages among live cells in WT nerve and in PNFs from Vehicle-, MEKi-, or SHP2i-treated mice. **(D)** UMAP visualization of CD45⁺ immune cells from Vehicle-, MEKi (PD0325901)-, and SHP2i (RMC-4550)-treated PNFs, downsampled to an equal number of CD45⁺ cells (9,506 per condition). **(E)** Quantification of CD163⁺ and CD163⁻ macrophages as a percentage of CD45⁺ immune cells in WT nerve and PNFs. Statistical significance is indicated (*P < 0.05, **P < 0.01, ***P < 0.001, ****P < 0.0001). **(F)** Morphometric evaluation of Iba1+ macrophage morphology using principal component analysis (PCA). SHP2i (RMC-4550)-treated macrophages exhibit distinct clustering based on shape and size compared with Vehicle- or MEKi (PD0325901)-treated samples. **(G)** Stacked bar plots of immune cluster frequencies in PNFs from Vehicle-, MEKi (PD0325901)-, and SHP2i (RMC-4550)-treated mice, showing a unique reduction in specific populations with SHP2 inhibition.

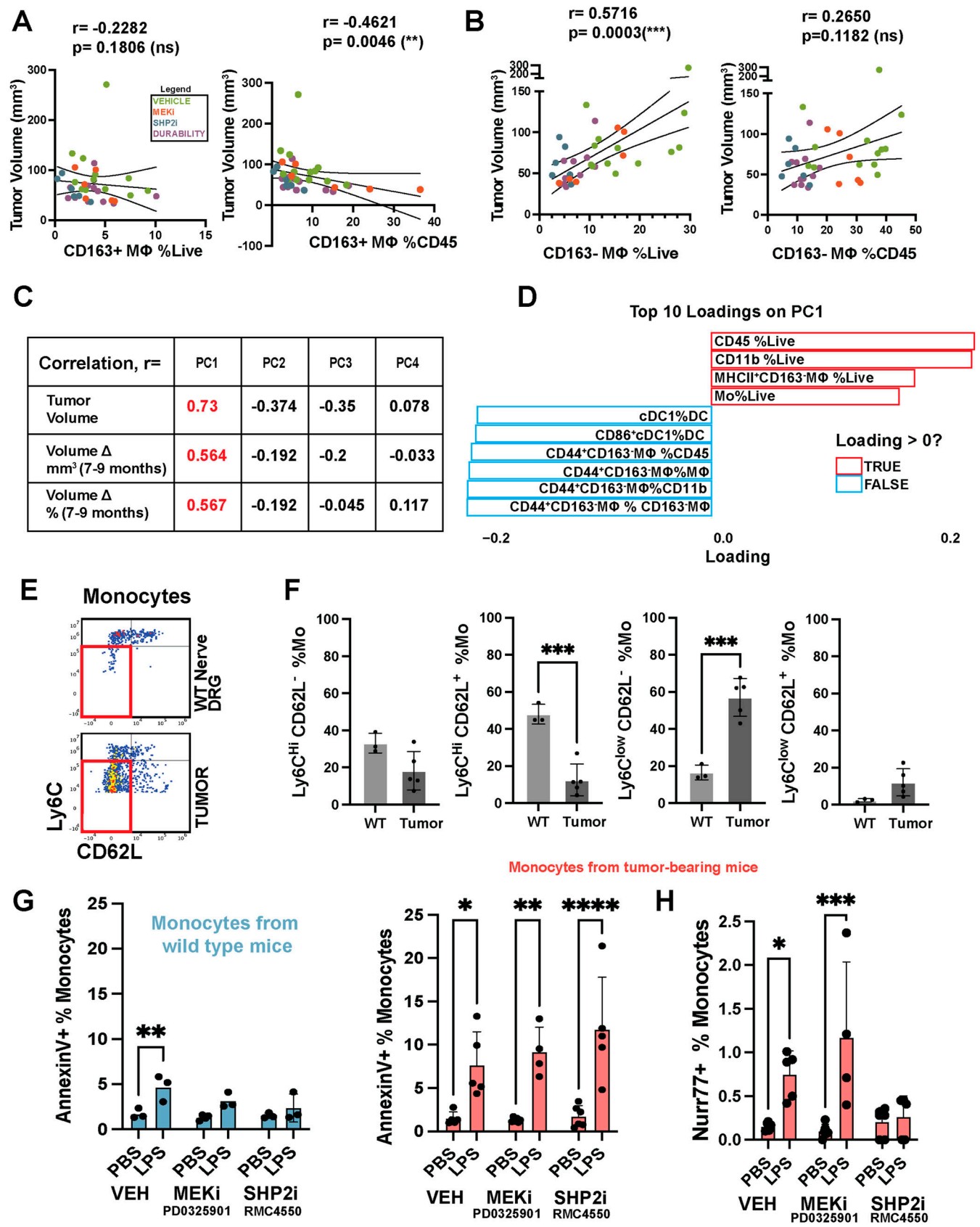

night), because of diurnal shifts in monocyte phenotypes. Indeed, treatment at night for 55 d, followed by 5 d of SHP2 daytime administration, substantially reversed drug effects observed when SHP2 inhibitor was administered for 55 d during the daytime (Fig 7B and C).

Monocytes derived from bone marrow circulate in the blood and move into tissues, including tumors (Fig 7D). To test whether circadian patterns of monocyte production, release, and/or differentiation are altered in the DhhCre;Nf1fl/fl tumor model, we analyzed blood cells collected during day or night in tumor-bearing mice, compared with WT non–tumor-bearing mice. Significant perturbations in circulating blood monocytes were present in tumor-bearing mice (Figs 7C and S7C). Blood monocytes lose expression of the marker CD62L as they age and patrol. Consistent with the presence of chronic inflammation in tumor-bearing mice, short-lived Ly6C$^{high}$;CD62L$^+$ were elevated in circulation in tumor-bearing mice, whereas Ly6C$^{low}$;CD62L$^-$ were lower in circulation. In WT mice, Ly6C$^{low}$;CD62L$^-$ monocytes are normally reduced in circulation at night consistent with prior reports of circadian regulation of monocyte subsets (Nguyen et al, 2013); this diurnal pattern was lost in tumor-bearing mice. Tumor-bearing mice treated with SHP2 inhibitor at night further increased the Ly6C$^{low}$;CD62L$^-$ monocytes in blood (Figs 7D and S7C). This was accompanied by a significant increase in Ly6C$^{hi}$CD62L$^-$ blood monocytes when SHP2 inhibition was administered at night (Fig S7D). Gating strategy for identification of monocyte and T-cell subsets in blood is shown in Fig S7B.

Importantly, as in circulation, monocytes and MDSC increased in abundance in tumors from mice treated with the SHP2 inhibitor at night (Fig 7E). There was also a shift in monocyte phenotype, with a change in the proportion of Ly6C$^{low}$;CD62L- monocytes (Fig 7F). This nighttime increase extended to both monocytes and NuMDSCs within the tumor, alongside significant alterations in Ly6C$^{hi}$ and Ly6Clow monocyte subset composition (Fig 7G and H). Macrophages and CD163$^-$ macrophages expressing the activation marker CD86 also decreased in tumors following SHP2 inhibitor treatment at night, correlating, as for anti-PD1 combination therapy, to response failure (Fig S7F); the frequency of CD4 and CD8 T cells remained unchanged day or night in treated mice (Fig S7E). We conclude that response to SHP2 inhibitor is reversed by factors that modulate myeloid cell phenotypes. The importance of myeloid cells in tumor response is emphasized by our finding that increased abundance of monocytes and macrophages correlates with tumor growth.

## Discussion

We identify the SHP2 phosphatase as a novel potential therapeutic target in PNF and use single-cell RNA-seq and multiparametric flow cytometry to define myeloid cell phenotypes in neurofibromas. MEK and SHP2 inhibition each reduced tumor volume and depleted recruited macrophages; MEK inhibition also dramatically reduced macrophages and dendritic cells in PNF from individuals with human NF1. We also found that monocyte trafficking and immune cell activation are abnormal in tumor-bearing mice; thus, tumor-bearing mice show disrupted diurnal patterns of circulating monocytes, and administering SHP2 inhibitor during the day (when these nocturnal animals are at rest) maximizes therapeutic efficacy. Conversely, giving the drug at night or combining it with the immune checkpoint inhibitor anti-PD1 reverses or negates SHP2 inhibitor–mediated tumor shrinkage by changing monocyte and macrophage phenotypes rather than T-cell frequency. Together, these results highlight how SHP2-regulated monocyte activation and circadian influences on immune cell trafficking critically shape neurofibroma growth and treatment response.

We showed that cobimetinib, like the previously tested MEK inhibitors PD0325901 (Jessen et al, 2013) and selumetinib (Dombi et al, 2016), shrinks murine neurofibromas. MEK and SHP2 inhibition similarly shrank tumors in our preclinical studies, suggesting that SHP2 inhibition might act as an alternative for MEK inhibition in neurofibroma therapy. We used the MEK inhibitor PD0325901 for most experiments in this study because it has been used clinically in neurofibroma (Weiss et al, 2021). Both cobimetinib and PD0325901 are dosed daily, as is RMC-4550. Early results suggest an acceptable tolerability profile of SHP2 inhibitors in single-agent clinical trials (Ou et al, 2020; Brana et al, 2021; Drilon et al, 2023). Given that in our preclinical tests, neither inhibition of MEK nor inhibition of SHP2 was durable, as a single-agent SHP2 inhibition is unlikely to be more effective than MEK inhibition. Nevertheless, because in mouse and human MEK inhibition causes only an average of 20% tumor volume reduction, and because 30% of individuals fail to respond to MEK inhibition (Gross et al, 2020), improvement over or alternatives to current MEK inhibition therapy may be useful clinically.

---

**Figure 5.  Principal component analysis (PCA) and correlation of macrophage/monocyte subsets with tumor volume.**
**(A)** Scatter plots of tumor volume (mm$^3$) versus the percentage of CD163+ macrophages among live cells (left) or CD45$^+$ cells (right). Pearson's correlation coefficient (r) and *P*-values are indicated; each point represents an individual tumor, color-coded by the treatment group. **(B)** Scatter plots of tumor volume (mm$^3$) versus the percentage of CD163$^-$ macrophages among live cells (left) or CD45$^+$ cells (right), with r and *P*-values indicated. Each dot represents an individual tumor, color-coded by the treatment group: Vehicle (green), MEKi (selumetinib) (red), SHP2 inhibitor (RMC-4550) (blue), and durability (purple; treated with RMC-4550 for 30 d followed by 30-d off treatment). Pearson's correlation coefficient (r) and *P*-values are shown. **(C)** Table summarizing Pearson's correlation coefficients (r) of tumor volume and volume changes (7–9 mo) with principal components (PC1–PC4). Values in red denote stronger correlations. **(D)** Bar chart of the top 10 loadings on PC1 in the PCA. Positive loadings (red bars) and negative loadings (blue bars) indicate the immune parameters most strongly contributing to PC1. **(E)** Flow cytometry plots illustrating monocyte gating (Mo) in WT dorsal root ganglia (DRG) versus tumor. Subsets are defined by Ly6C and CD62L expression (e.g., Ly6C$^{hi}$ CD62L+, Ly6C$^{low}$ CD62L–, etc.). **(F)** Quantification of monocyte subsets in WT DRG and tumor, expressed as a percentage of total monocytes. Data represent the mean ± SEM, with statistical significance indicated (*P < 0.05, **P < 0.01, ***P < 0.001). **(G)** Annexin V staining of monocytes gated from whole bone marrow from WT (left) or tumor-bearing (right) mice treated ex vivo with vehicle (VEH), MEK inhibitor (MEKi, PD0325901), or SHP2 inhibitor (SHP2i, RMC-4550), with or without LPS stimulation. Bars show the percentage of Annexin V+ monocytes. Data represent the mean ± SEM, with significance indicated (*P < 0.05, **P < 0.01, ***P < 0.001, ****P < 0.0001). **(G, H)** Nur77 expression in monocytes gated from the same ex vivo experiments as in (G), highlighting differences in expression levels among treatment groups with and without LPS stimulation.

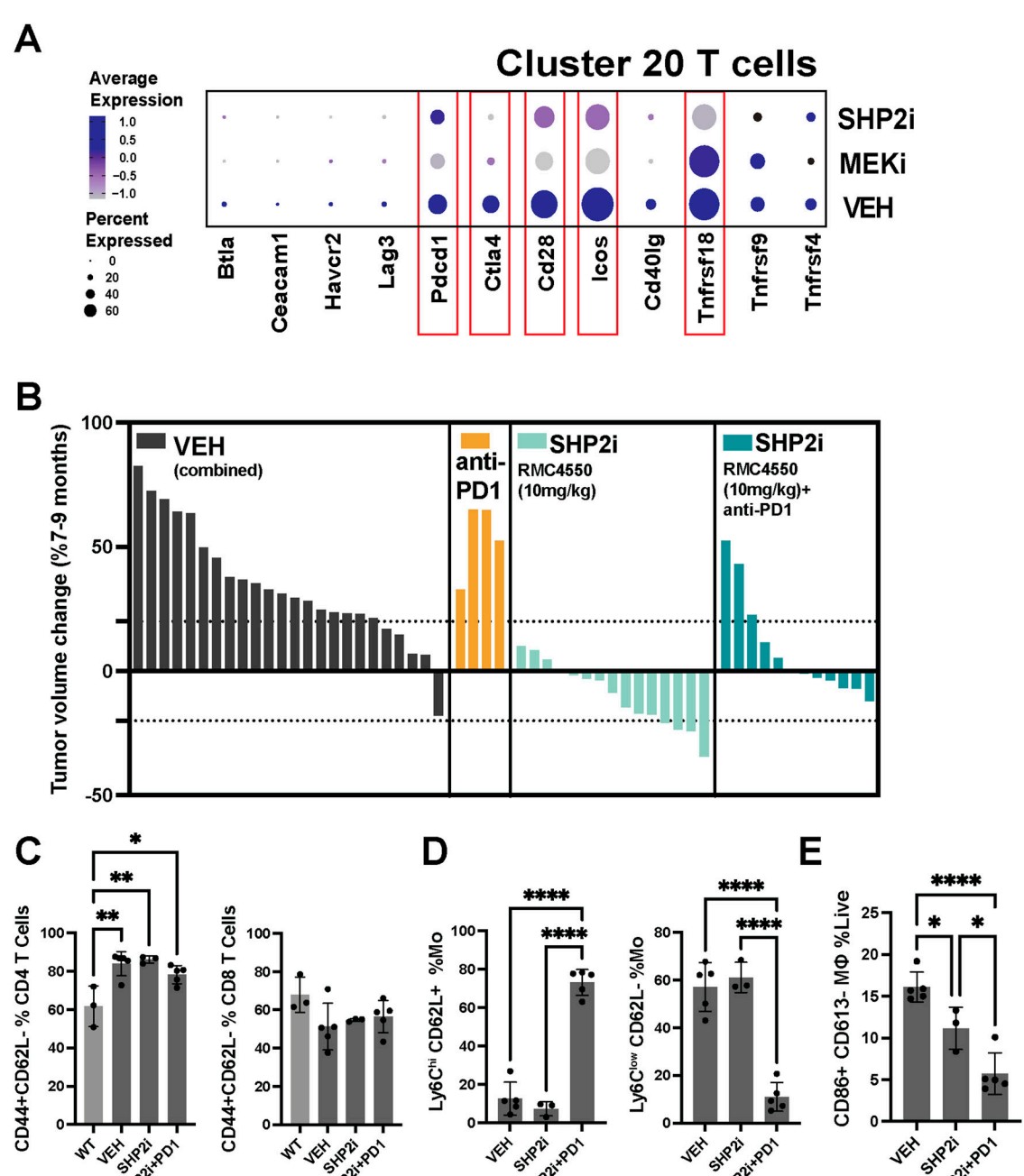

**Figure 6. Combining SHP2 inhibition with PD-1 blockade modulates T cells and monocyte populations in plexiform neurofibromas (PNFs) and tumors grown.**
**(A)** Dot plot showing the expression of key activation and checkpoint markers (*Btla, Ceacam1, Havcr2, Lag3, Pdcd1, Ctla4, Cd28, Icos, Cd40lg, Tnfrsf9, Tnfrsf4, Tnfrsf18*) in Cluster 20 T cells. The size of each dot indicates the percentage of cells expressing that marker, whereas color intensity reflects the average expression level. Treatments include Vehicle (VEH), MEKi (PD0325901), and SHP2i (RMC4550). **(B)** Waterfall plots displaying tumor volume changes (% change from 7 to 9 mo) in mice treated with VEH (combined), anti-PD1, SHP2 inhibitor (SHP2i, RMC-4550 at 10 mg/kg), or SHP2i plus anti-PD1. Each bar represents an individual tumor. **(C)** Bar graphs showing the frequency of activated T cells (CD44+CD62L−) among CD4+ (left) and CD8+ (right) T cells in WT nerve or PNFs from VEH-treated, SHP2i-treated, and SHP2i plus anti-PD1–treated groups. **(D)** Quantification of monocyte and macrophage subsets, including Ly6Chi CD62L+ monocytes, Ly6Clow CD62L+ monocytes (both as %Mo), and macrophages (MΦ) as a percentage of total CD45+ cells. **(E)** Quantification of CD86+CD163− macrophages (MΦ) as a percentage of live immune cells in PNFs from VEH-treated, SHP2i-treated, and SHP2i plus anti-PD1–treated groups. Data are shown as the mean ± SEM, with significant differences indicated (*$P < 0.05$, **$P < 0.01$, ***$P < 0.001$, ****$P < 0.0001$).

Combining MEK and SHP2 inhibition overcomes resistance to single agents in some settings (Wang et al, 2020; Drilon et al, 2023). However, our analysis did not reveal beneficial effects of combination therapy in PNF. Rather, each agent caused tumor shrinkage in about 70% of mice, and the combination was like SHP2 alone in the extent of inhibition of pERK in tumors. Given the

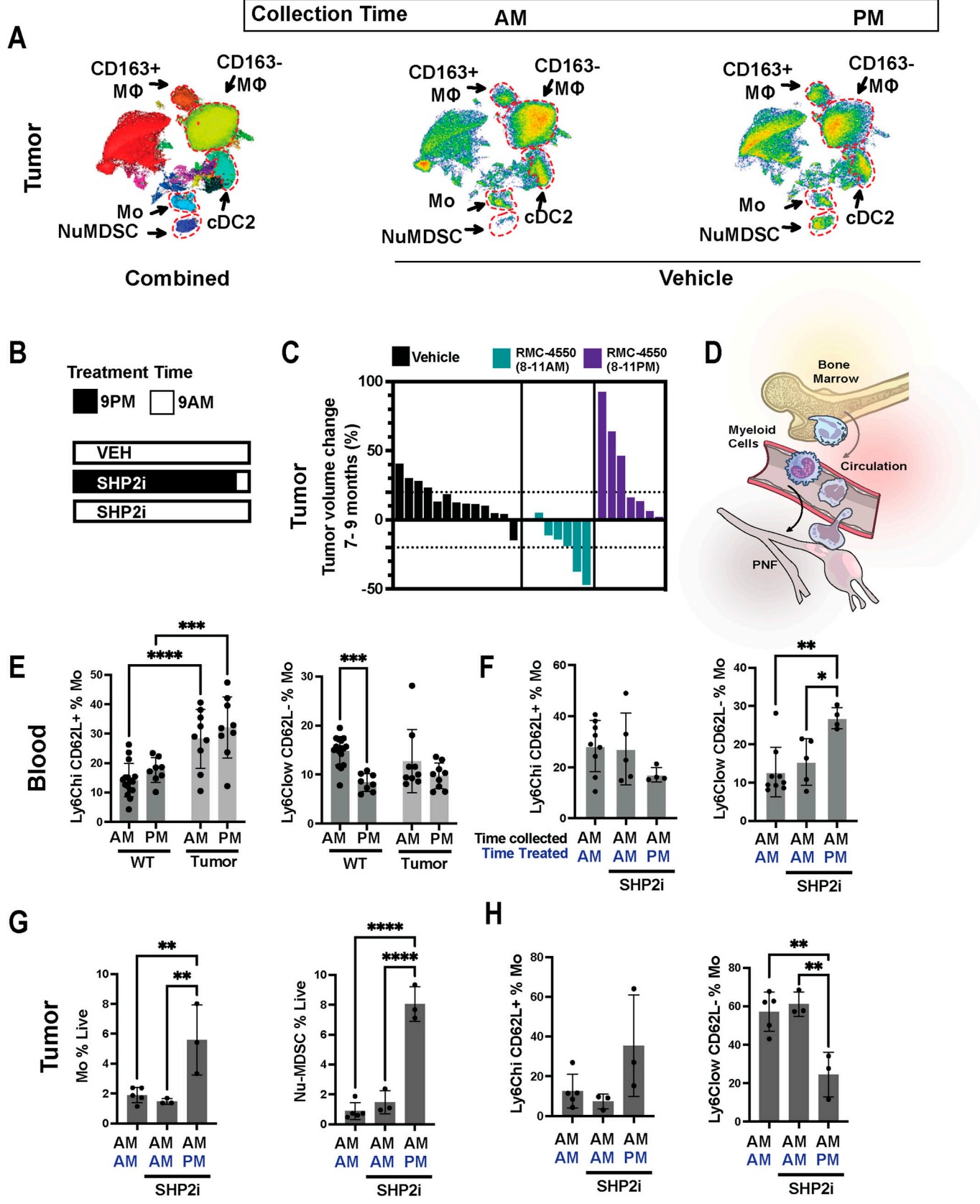

concordance between extent of response and percentage of responses to MEK inhibition in the murine model used here (Jessen et al, 2013; Dombi et al, 2016) and in humans with inoperable NF1 PNFs (Dombi et al, 2016; Gross et al, 2020), this combination of therapies is not predicted to provide improved therapeutic benefit in this setting.

We identified transcriptional effects of MEK and SHP2 inhibitors on pathways including epithelial–mesenchymal transition, NFKB signaling, and RAS signaling, the same pathways that are activated in neurofibroma versus control nerve/DRG (Kershner et al, 2022). Also, matrix accumulation is significant in neurofibroma, and our finding that MEK inhibition blocks transcription of numerous matrix components by Schwann cells, which may contribute to tumor shrinkage effects, is consistent with a recent publication (Jiang et al, 2023). The similar effects of the two inhibitors on gene expression in tumor glial cell populations likely reflect on-target pathway effects in tumor cells regardless of their *Nf1* status, as cells with *Nf1* knockout are a minority in this model (Wu et al, 2008). More detailed analysis of inhibitor effects on glial cell populations remains of interest. MAPK pathway blockade alters cytokine/chemokine/growth factor release from tumor cells, known to increase in *Nf1*–/– Schwann cells (Yang et al, 2003), and reduced factor release combined with effects on immune cells may contribute to the overall effects on immune cells, and in principle differ between inhibitors, contributing to effects on immune cells in tumor-bearing mice.

MEK inhibition and SHP2 inhibition each compressed the PNF myeloid compartment. SHP2 inhibition decreased nonclassical monocytes in circulation and reduced numbers of CD163-negative macrophages in tumors. In normal DRG, CD163⁻ macrophages rapidly turn over, replenished by blood monocytes (Lund et al, 2024), and these CD163⁻ cells are the macrophages that expand in neurofibromas. Both inhibitors also up-regulated some immune modulators in immune cells. These included *Il10*, *Ccl2*, and *Cxcl2*, which are also up-regulated in response to SHP2 inhibition in melanoma (Tang et al, 2022; Christofides et al, 2023). Yet, other effects of SHP2 inhibition differed from those of MEK inhibition. SHP2 inhibition, but not MEK inhibition, altered Nur77 expression on monocytes from tumor-bearing mice and changed aspects of MDSC transcriptional profiles. Of note, re-shaping of MDSC by SHP2 inhibitor was also reported in a melanoma model

(Christofides et al, 2023), and these results support further functional study of the role of MDSC in PNF tumorigenesis. Finally, SHP2 inhibition homogenized macrophage populations, as readout transcriptionally, by flow cytometry and by macrophage morphology. Our findings are consistent with the idea that SHP2 inhibitors alter the neurofibroma microenvironment via direct effects on tumor cells and through direct effects on immune cells in the bone marrow and in circulation before they are recruited into tumors. This idea is consistent with the finding that ablation of *Ptpn11*/SHP2 in myeloid cells is protective in inflammation-induced colitis, a precancer setting, and in transformation to colon cancer (Xiao et al, 2018), and consistent with effects described in a diverse set of cancers (Tang et al, 2022; Quintana et al, 2020; Christofides et al, 2023; reviewed in Sodir et al [2023]). Even when tumors are not present, as in the setting of chronic inflammation, there is a dominant effect of SHP2 inhibition on immune cells (Barkal et al, 2019). Thus, in the benign tumors we have studied, there are tumor cell–intrinsic effects of SHP2 inhibition and effects on monocytes, monocyte differentiation, and altered phenotypes of myeloid-derived cells.

In PNF, as in several other precancerous model systems, T-cell activation promotes tumor growth (Roberts et al, 2007; Endig et al, 2016; Fletcher et al, 2019; Pundavela et al, 2024). In neurofibroma, SHP2 inhibition not only reduced tumor myeloid cells but also increased tumor T cells as a percentage of tumor immune cells, as in a GEM model of non–small-cell lung cancer (NSCLC) (Tang et al, 2022). SHP2 inhibition may also directly affect PNF lymphocytes. Arguing for a direct effect, conditional ablation of *Ptpn11*/SHP2 in CD4 T cells increased tumor progression in a melanocyte model (Zhang et al, 2013). Additional effects of SHP2 inhibition on T cells may be indirect. For example, knockout of *Ptpn11* in myeloid cells increased T-cell activity (Christofides et al, 2023), and SHP2 inhibition depleted pro-tumor macrophages (Quintana et al, 2020). Drug therapy simultaneously affects tumor and immune cells, so both direct and indirect effects of SHP2 inhibition may be relevant.

A major finding of this study is that dosing SHP2 at different times of day affects drug response. Circadian changes in immune cell activation and circulation have been noted for many years, and recently used to guide drug dosing and immunotherapy (Mok et al, 2024; Wang et al, 2024). We found that differences in immune cell

**Figure 7. Day versus night administration of the SHP2 inhibitor modulates myeloid cell dynamics and tumor regression.**
**(A)** UMAP analysis of tumor-infiltrating myeloid cells under AM versus PM collection times, highlighting differences in CD163+ macrophages (MΦ), CD163⁻ MΦ, monocytes (Mo), conventional dendritic cells (cDC2), and neutrophil-like MDSCs (NuMDSC). "Combined" shows all cells merged, whereas "Vehicle" illustrates cells from untreated controls. **(B)** Schematic illustrating the experimental setup. Mice were treated with Vehicle (VEH) or SHP2 inhibitor (SHP2i) at either 9 AM or 9 PM, with all samples collected at 9 AM to assess the time-of-day effects on myeloid populations. **(C)** Waterfall plot illustrating the percent change in tumor volume from 7 to 9 mo in mice treated with Vehicle (black bars) or SHP2 inhibitor (RMC-4550) administered from 8 to 11 AM (green bars) or 8 to 11 PM (purple bars). Each bar represents an individual mouse; positive values indicate growth, and negative values indicate shrinkage. Bar graphs showing the frequencies of Ly6C^hiCD62L+ (classical) and Ly6C^lowCD62L- (nonclassical) monocyte subsets in the blood of WT and tumor-bearing mice, collected during 9 AM or 9 PM. **(D)** Schematic depicting myeloid cell trafficking. Bone marrow–derived myeloid cells circulate through the bloodstream and infiltrate plexiform neurofibromas. **(E)** Bar graphs compare the frequencies of distinct monocyte subsets, including Ly6C^hiCD62L+ (classical) and Ly6C^lowCD62L– (nonclassical), among monocytes in blood. **(F)** Frequencies of Ly6C^hiCD62L+ and Ly6C^lowCD62L– monocyte subsets among blood monocytes in vehicle- or SHP2 inhibitor–treated mice, comparing collection time (AM versus PM) and treatment time (AM versus PM). **(G)** Quantification of monocytes (Mo, % live) and neutrophil-like MDSCs (NuMDSCs, % live) in tumors from mice treated with SHP2i at either 9 AM or 9 PM, with vehicle-treated mice as a control (all treated and collected at 9 AM). **(H)** Quantification of tumor monocyte subsets (Ly6C^hiCD62L+ and Ly6C^lowCD62L–) as percentages of total monocytes (%Mo) in SHP2i-treated mice (treated at 9 AM or 9 PM) and vehicle-treated mice (treated and collected at 9 AM). Additional comparisons of monocyte/granulocyte markers in bone marrow collected AM versus PM, including Ly6C^hiCD62L⁺ monocytes and CD62L expression on CD11b+Ly6C^hiLy6G⁺ cells. Bars show the mean ± SEM, with significance indicated (*P < 0.05, **P < 0.01). Data represent the mean ± SEM; statistical significance is indicated (e.g., **P < 0.01, ***P < 0.001, ****P < 0.0001).

populations during day and night occur at the site of immune cell production, in circulation, and in tumors in tumor-bearing mice. This information is consistent with findings from other disease states, in which diurnal rhythms of immune cell trafficking and differentiation are disrupted (Mok et al, 2024). Of note, recent publications show that circulating immune cells in individuals with NF1 and neurofibromas also differ from immune cell controls (Nobeyama et al, 2022). It will be of interest to determine whether immune cells also differ in individuals with NF1 but who do not have tumors.

In conclusion, SHP2 inhibition has single-agent activity in a predictive preclinical model of neurofibroma and differs from MEK inhibition in its modulation of the tumor immune milieu. These findings suggest that small molecule therapy by SHP2 inhibition acts as an immunotherapy, which may be especially useful in neurofibroma where traditional antibody-based immunotherapy and its associated toxicities may not be tolerable. Finally, special consideration should be given to the timing of SHP2 administration in clinical settings.

### Limitations

Although our findings strongly support SHP2 inhibition as a therapeutic approach for PNFs, there are important limitations. First, SHP2 itself has broad, context-dependent effects on multiple pathways that regulate monocyte production, trafficking, and activation. Thus, it remains unclear which compartment—bone marrow, blood, or tumor—is most critical to these effects. Moreover, fibroblasts, which have been shown to regulate the circadian infiltration of T cells (Mok et al, 2024), may also influence how SHP2 inhibition alters immune cell populations. Likewise, whether the circadian effects of SHP2 inhibition stem primarily from changes in monocyte infiltration, production, trafficking, or maturation, or whether they are secondary to T-cell interactions, cannot be definitively concluded from our current data. Given the different ways MEK and SHP2 inhibition modulate the immune system, we predict that SHP2 inhibitors—but not MEK inhibitors—will be significantly affected by the combinations we tested, such as anti-PD1 therapy or circadian-based administration. We emphasize the complexity of targeting SHP2, which indirectly and directly influences multiple cell types and signaling cascades, highlighting the need for further studies to clarify how these pathways intersect.

## Materials and Methods

### Mice and husbandry

The experimental protocols followed in this study received approval from the Institutional Animal Care and Use Committee at Cincinnati Children's Hospital Medical Center. We maintained the mice in an environment with controlled temperature and humidity, ensuring they had continuous access to food and water. The facility's light–dark cycle was set to 12 h. The mouse strains used in this research were Nf1fl/fl and DhhCre (Wu et al, 2012). For breeding

purposes, we kept the DhhCre allele in males and crossed DhhCre; Nf1fl/+ mice with Nf1fl/fl mice to produce mice with tumors. The mice used in this study were all bred on a C57BL/6 genetic background. For the identification of Nf1 genotypes, we employed specific oligonucleotides: the sequence CTT CAG ACT GAT TGT TGT ACC TGA and ACC TCT CTA GCC TCA GGA ATG A for detecting the WT allele, and TGA TTC CCA CTT TGT GGT TCT AAG for the targeted allele. Genotyping of DhhCre was carried out using the forward primer ACCCTGTTACGTATA GCCGA and the reverse primer CTCCGGTATTGA AACTCCAG. In addition, GFP genotyping was conducted using the forward ACGTAAACGGCCACAAGTTCA and reverse GCTGTTGTAGTTGTA CTCCAGGT primer.

### Compound administration

Mice were acclimated to handling and dosed once daily. Two MEK inhibitors were used: cobimetinib (GDC-0973) (a kind gift from Genentech) at 5 mg/kg (for experiments in Fig 1) and mirdametinib (PD0325901), purchased from Selleck (Cat #S1036), at 1.5 mg/kg for all other studies. Both MEK inhibitors were prepared in a 0.5% (wt/vol) methylcellulose (Cat #M0262-100G; MilliporeSigma) solution with 0.2% (vol/vol) polysorbate 80 (Tween-80, Cat #BP338-500; Thermo Fisher Scientific) and administered via oral gavage. The SHP2 inhibitor RMC-4550, provided by Revolution Medicines under a former collaboration agreement with Sanofi, was prepared in a 50 nM sodium citrate (Cat #71497-250G; MilliporeSigma) buffer (pH 4.0 ± 0.1) with 0.5% (vol/vol) polysorbate 80 (Tween-80), and administered at either 10 mg/kg or 30 mg/kg once daily via oral gavage. The RMP1-14 monoclonal antibody anti-mouse PD-1 (CD279), procured from BioXCell (Cat #BE0146), was diluted in sterile PBS and administered via intraperitoneal (i.p.) injection once a week at a concentration of 12.5 mg/kg. Controls were either oral gavage or i.p. injection with an empty vehicle, as appropriate for each experiment.

*Magnetic resonance imaging and analysis* was performed in anesthetized mice as described using 7T Bruker Biospec System to acquire images in three planes to position the 3D volume (Wu et al, 2012). We calculated tumor volume from the area of graphic outlines and MRI slice thickness. To derive *P*-values, we conducted a random-effects model analysis on the log-transformed tumor volume data using the SAS mixed procedure.

### IHC

We anesthetized mice, then perfused them with ice-cold PBS by cardiac puncture, and then rapidly dissected tissues that were fixed in 10% freshly prepared PFA (for 24 h) and embedded in paraffin. Sections (4–5 μm thick) were deparaffinized, rehydrated, and subjected to antigen retrieval using standard protocols. We blocked sections with 5% normal goat serum (Cat #50-062Z; Thermo Fisher Scientific—Invitrogen). Primary rabbit anti-Ki67 (1:300, Cat #12-201; Cell Signaling) and anti-phosphorylated extracellular signal–regulated kinase1/2 (pERK) (1:300, Cat #7340; Cell Signaling) were applied, followed by incubation with species-specific biotinylated secondary antibodies (Goat α-Rb) (Cat #BA-1000; Vector Labs or Cat #31671; Thermo Fisher Scientific). Detection was performed using a streptavidin–biotin–peroxidase complex

method (Cat #PK-6101; Vector Labs, VECTASTAIN), and visualization was achieved with 3,3′-diaminobenzidine (DAB) substrate (Cat #SK-4100; Vector Labs). Counterstaining was done with hematoxylin (Cat #6765001; Thermo Fisher Scientific) for Ki67 slides only. Coverslips were mounted using Histomount (Cat #HS-103; National Diagnostics). Four images per tissue were captured and used for analysis. Quantification of Ki67-positive cells was performed by hand, and pERK staining intensity was assessed using relative values. Statistical analysis assessed differences between experimental groups, and positive and negative controls were included.

### Western blots

Tumor or lung tissue lysates were prepared in RIPA buffer (#89900; Thermo Fisher Scientific) with added protease (#P8340; Sigma-Aldrich) and phosphatase inhibitors (#P5726; Sigma-Aldrich), and protein concentrations were determined using the BCA kit (#500-0111; Bio-Rad). Equal amounts of protein were loaded onto 4–20% gels (#456-1096; Bio-Rad Mini-PROTEAN TGX), transferred onto nitrocellulose membranes (#162-0115; Bio-Rad), and separated by electrophoresis. The membrane was then blocked with 5% nonfat (Blotting-Grade #170-6404; Bio-Rad) in PBS/Tween (0.1%). Western blot analysis used antibodies from Cell Signaling pERK1/2 (Cat #3179S; 1:2,000) and total ERK1/2 (Cat #4695S; 1:2,000). Anti-pERK was detected overnight at 4°C; after washing, membranes were incubated with horseradish peroxidase–conjugated secondary antibodies (#7074S; CST) and ECL detection substrate (SuperSignal West Pico #34580; Thermo Fisher Scientific). After stripping (Restore #21059; Thermo Fisher Scientific) the membrane, anti-total ERK was applied, and detection was carried out overnight at 4°C as above.

### Flow cytometry

After euthanasia, mice were briefly perfused intracardially with ice-cold PBS and tissues removed and stored in Opti-MEM with GlutaMAX (#31985-070; Thermo Fisher Scientific) additives at 4°C for 3–36 h. For tissue dissociation, we prepared fresh enzyme mix (1 ml/sample/1.5 ml tube). The enzyme mix consisted of 100 $\mu$l collagenase A (100 mg/ml) (# 10103586001; Sigma-Aldrich), 100 $\mu$l collagenase type 4 (100 mg/ml) (# LS004188; Worthington Biochemical), 40 $\mu$l soybean trypsin inhibitor (100 $\mu$g/ml) (# T9128; Sigma-Aldrich), 10 $\mu$l DNase I (250 U) (# DN25; Sigma-Aldrich), 5 $\mu$l 1 M CaCl$_2$ (5 mM final concentration) (R21457; Thermo Fisher Scientific), and 745 $\mu$l complete RPMI media (# 11875-093; Thermo Fisher Scientific [Gibco]). A shaker was warmed to 37°C. Tissue was centrifuged for 5 min at 300$g$ at 4°C; most media were removed and minced into small pieces (~1 mm$^3$) and placed into the enzyme mix. The tube was then placed on the shaker at 37°C for 30 min and transferred to a 50-ml conical tube, and enzymatic dissociation was halted by adding 30 ml RPMI + 1% FBS. After gentle mixing and centrifugation for 5 min at 300$g$ at 4°C, the supernatant was partially removed, leaving 3–5 ml. After adding ~25 ml RPMI + 1% FBS, filtering the cell suspension through 100- and 70-$\mu$m cell strainers for a single-cell solution, centrifugation, removal of supernatant, and resuspending the pellet in ~1 ml for subsequent cell counting without dilution, we counted cells. One million cells

from each sample were transferred into standard 5-ml flow tubes (352235; Corning) and washed twice with 1 ml of PBS to eliminate residual serum. Subsequently, a Live/Dead marker (L23105; Thermo Fisher Scientific) was used to exclude nonviable cells. To identify apoptotic cells, Apotracker Green (427402; BioLegend) was used before subsequent staining. After Live/Dead exclusion, cells were incubated for 60 min at room temperature in a 100 $\mu$l antibody mixture (see Table S2). After the incubation period, unbound antibodies were removed by washing the cells with 1.5 ml of PBS twice. The cells were then fixed with freshly prepared 2% PFA in PBS for 30 min, washed and resuspended in 350 $\mu$l in PBS for immediate analysis, or stored in 500 $\mu$l Cyto-Last Buffer (Cat #422501; BioLegend), and then, cells were analyzed on a Cytek Aurora flow cytometer with a 5-laser configuration (Cytek) within 10 d.

### Flow analyses

We used UltraComp beads (Cat #01-2222-41; Thermo Fisher Scientific) and Live/Dead ArC amine-reactive compensation bead kits (Cat #A10346; Thermo Fisher Scientific) for single color controls. Each tumor or bone marrow sample was run together with its own negative control (unstained) to control for endogenous fluorescence. Tumors from 3 to 11 mice were analyzed for each treatment, as designated for each experiment. We used data from three individual antibody panels (Fig S3G). Fcs files from individual mice were downsampled and concatenated to include the same number of events for each tumor, using FlowJo software. UMAP analysis was performed in FlowJo with default settings of nearest neighbors of 15 and minimum distance. Data were scaled before generating UMAP or FlowSOM clustering and analysis. Cluster explorer was used to generate expression profiles with a cutoff of 99% ILE (outliers were removed), and means of scaled values were used to generate plots.

### Single-cell analysis

Mouse: Within 5 min after euthanasia, paraspinal tumors were excised and placed into ice-cold L-15 medium (MT-10-045-CV; Thermo Fisher Scientific). Tissue was cut into 1-mm$^3$ pieces and placed into dissociation medium in 50-ml tubes containing 20 ml L-15 medium (# 11415064; Thermo Fisher Scientific [Gibco]), 10 mg collagenase type I (LS004196; Worthington Biochemical), and 50 mg Dispase II (04942078001; Sigma-Aldrich) in a 37°C incubator for 2 h, with 170 rpm shaking. The dissociation process was stopped by adding 30 ml DMEM (11-965-1188; Thermo Fisher Scientific) containing 10% FBS. Samples were centrifuged at 800$g$ for 5 min at 4°C. Supernatants were removed, and cell pellets were resuspended in 50 ml of 1× PBS containing 0.1% BSA. Suspensions were sequentially filtered with 70-$\mu$m (08-771-2; Thermo Fisher Scientific), 40-$\mu$m (1181X52; Thomas Scientific), and 20-$\mu$m cell strainers (43-50020-03; pluriSelect) and centrifuged at 600$g$ for 5 min at 4°C. Pellets were resuspended in 100 $\mu$l of 1× PBS/0.1% BSA. Trypan blue–negative viable cells were counted to determine the cell concentration for droplet-based scRNA-seq. The 10X Genomics scRNA-seq (3′ v3.1 chemistry) libraries were aligned to the mm10 mouse genome using the Cell Ranger pipeline (version 6.0) with default parameters. Downstream analyses were performed using Seurat (version 4.1.1,

https://satijalab.org/seurat/). Data normalization, dimensionality reduction, clustering, and integration were performed using the standard Seurat v4 integration workflow as previously described (Kershner et al, 2022). For clustering functions, the dimension was set to 1:30 and resolution to 0.6. During sample integration, default parameters were used in Seurat's FindIntegrationAnchors function, including setting anchor features to 2,000. Cell-type cluster annotation was performed by comparing the top 50 cluster markers identified with the FindAllMarkers function in Seurat to published literature. Differential expression analyses were performed using Seurat's FindMarkers function and setting logfc.threshold = 1. Preliminary cluster annotation of tumors contains cell types discovered in PN (Kershner et al, 2022). There was no significant difference between untreated and vehicle-treated tumors, so we merged these controls for further analysis, and compared the percentage of cells in each annotated cell cluster with treatment groups. The scRNA-seq data for 7-mo-old normal control (DRG/nerve) samples were extracted from Kershner et al (2022). Here, 7-mo-old tumor samples correspond to "Vehicle" samples in the current study. For each cell cluster, differentially expressed genes (DEGs) were predicted between MEKi (or SHP2i) and Vehicle groups, using Seurat's FindMarkers function. DEGs were considered significant DEGs if their |fold| > 1.25x their |fold| > 1.25x and adjusted $P < 0.05$. Significant up- and down-DEGs were queried in Enrichr (https://maayanlab.cloud/Enrichr). Enrichment tests were performed against the MSigDB HALLMARK gene sets using combined (i.e., up- and down-) DEGs as a query set. Any significantly enriched gene (|fold| > 1.25x; Adj. $P < 0.05$) sets implied gene set/pathway "perturbation." When the transformed score was over 6, we reset it to 6 for color scaling (0 = weakest; 6 = strongest enrichment). Raw data, including FASTQs and count matrices, are available in the Gene Expression Omnibus database (GEO GSE181985).

### Single-cell sequencing for human subjects treated with the MEKi selumetinib

Informed consent was obtained from all patients, and treatment was conducted on ClinicalTrials.gov identifier: NCT02407405 (PMID 39762421). The collection of patient tumor samples, procedures for single-cell RNA sequencing, and data storage plans were approved by the National Cancer Institute Institutional Review Boards (NCT01109394; IRB identifier 10C0086). Procedures for the collection, banking, and process of patient NF1-associated peripheral nerve sheath tumors for single-cell RNA sequencing (scRNA-seq) using a 10x Genomics platform were described previously (Zhang et al, 2023). From each tumor sample, two capture lanes with a targeted 6,000 live cells per lane to be captured were processed on the Chromium Controller (10x Genomics). 10X v3 reagents were used for library preparation following the manufacturer's protocol. Sequencing of the cDNA library was performed on an Illumina NovaSeq 6000 sequencer, aiming to achieve at least 50,000 reads per cell. Data analysis was performed using Seurat v4, and marker genes of each cell cluster were identified with the Seurat function FindAllMarkers in the "RNA" assay. The top highly expressed and the most differentially expressed marker genes were used to manually annotate the cell types based on prior knowledge and their overlaps with the canonical markers. The generated sequencing data are deposited in dbGaP with the accession number phs003519.

### Bone marrow assay

Cells from male and female WT or *Nf1fl/fl;DhhCre*[+] tumor-bearing mouse femurs (6–12 mo of age) were extracted after storing for 24–36 h at 4°C after euthanasia. Bone marrow was flushed using Opti-MEM supplemented with GlutaMAX (Thermo Fisher Scientific), mechanically dissociated, filtered through a 40-micron cell strainer, and used immediately for flow cytometry or plated onto 12-well culture plates for 30 min for in vitro experiments using monocytes, for which nonadherent cells were washed away with warmed medium, and cells were treated with LPS (# 00-4976-03; Thermo Fisher Scientific) or vehicle and evaluated by flow cytometry.

### Statistical analysis

Tumor shrinkage analysis and random coefficients model analysis were completed as described in Wu et al (2012). All other statistical analyses were conducted in GraphPad Prism. Significance was set at $P \leq 0.05$. Statistical tests (one-way or two-way ANOVA, with Tukey's post hoc test) were as in figure legends. Error bars represent the SD unless otherwise noted.

### Ex vivo tumor assay

Untreated or vehicle-treated tumors were dissociated as described above. Single cells were washed twice with Opti-MEM with GlutaMAX, and we transferred $1 \times 10^6$ to wells of a round-bottom 96-well plate and incubated at 37°C with 5% $CO_2$ in 250 $\mu$l of Opti-MEM with GlutaMAX. Four hours later, we added RMC-4550 to 1 $\mu$M, PD0235901 to 1 $\mu$M, or vehicle (DMSO) control and incubated cells for 12 or 36 h. Plates were then placed on ice, and cells were labeled for flow cytometry analysis as described above. After fixation, we permeabilized cells with the True-Nuclear Transcription Factor Buffer Set (Cat #424401; BioLegend) and stained for cleaved caspase-3 (Asp175) (Cat #9661; Cell Signaling), followed by Donkey anti-Rabbit IgG (H+L) Highly Cross-Adsorbed Secondary Antibody, Alexa Fluor 647 (Cat #A-31573; Thermo Fisher Scientific). Samples were run on the Cytek Aurora and analyzed in FlowJo.

# Data Availability

The mouse single-cell RNA-sequencing data generated in this study have been deposited in the Gene Expression Omnibus (GEO) under accession number GSE181985. The human single-cell RNA-sequencing data from patients treated with the MEK inhibitor selumetinib are available in the Database of Genotypes and Phenotypes (dbGaP) under the accession number phs003519. Additional processed data supporting the findings of this study are available within the article and its supplementary information files.

# Supplementary Information

# Acknowledgements

This work was supported by DOD grant W81XWH-19-1-0816 (to N Ratner), R01 NS28840 (to N Ratner), and Revolution Medicines under a former collaboration agreement with Sanofi (to N Ratner). We thank Katherine Chaney Schaffer for assistance with dosing animals, and Jay Pundavela for assistance with dissections, and Ty Troutman and Daniel Lucas (CCHMC) for helpful critique of the article. We thank Bob Nichols and Elsa Quintana (Revolution Medicines) for encouragement, helpful discussions, and providing RMC-4550. We thank Mark Merchant (Genentech) for providing cobimetinib.

## Author Contributions

N Ahmari: conceptualization, investigation, visualization, and writing—original draft, review, and editing.
K Choi: formal analysis.
J Wu: investigation.
TA Rizvi: investigation.
M Jackson: investigation.
LJ Kershner: formal analysis.
M-O Kim: formal analysis.
X Zhang: formal analysis and investigation.
E Dombi: investigation.
J Shern: formal analysis.
DA Hildeman: conceptualization.
N Ratner: conceptualization, supervision, funding acquisition, visualization, project administration, and writing—original draft, review, and editing.

## Conflict of Interest Statement

The authors declare that they have no conflict of interest.

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
