## [Reviewer comments · Life Science Alliance]

Daytime SHP2 inhibitor dosing, when immune cell numbers are elevated, shrinks neurofibromas

Niousha Ahmari, Kwangmin Choi, Jianqiang Wu, Tilat Rizvi, Mark Jackson, Leah Kershner, Mi-Ok Kim, Xiyuan Zhang, Eva Dombi, Jack Shern, David Hildeman, and Nancy Ratner

DOI: <https://doi.org/10.26508/lsa.202503359>

Corresponding author(s): Nancy Ratner, Cincinnati Children's Hospital Medical Center

Review Timeline:

Submission Date:	2025-04-11
Editorial Decision:	2025-05-19
Revision Received:	2025-07-07
Editorial Decision:	2025-07-30
Revision Received:	2025-09-03
Accepted:	2025-09-08

Scientific Editor: Tim Fessenden

Transaction Report:

May 19, 2025

Re: Life Science Alliance manuscript #LSA-2025-03359-T

Dr. Nancy Ratner
Cincinnati Children's Hospital Medical Center
Department of Pediatrics
3333 Burnet Avenue
Cincinnati, Ohio 45229-0713

Dear Dr. Ratner,

Thank you for submitting your manuscript entitled "Immune modulation by time-of-day dosing determines the efficacy of SHP2 inhibition" to Life Science Alliance. The manuscript was assessed by expert reviewers, whose comments are appended to this letter. We invite you to submit a revised manuscript addressing the Reviewer comments.

As you will see, reviewers overall appreciated these findings on PNF tumors and novel strategies to target the myeloid populations that populate them. All reviewers made requests to improve the text, including for clarity and consistency in terminology of myeloid cell populations as well as drugs used. Reviewer 3 suggested ways to improve the overall flow and logic of the work. Finally all reviewers reviewers sought improvements to figure clarity.

The typical timeframe for revisions is three months. Please note that papers are generally considered through only one revision cycle, so strong support from the referees on the revised version is needed for acceptance. When submitting the revision, please include a letter addressing the reviewers' comments point by point.

Thank you for this interesting contribution to Life Science Alliance. We are looking forward to receiving your revised manuscript.

Sincerely,

-- By submitting a revision, you attest that you are aware of our payment policies found here: <https://www.life-science->

B. MANUSCRIPT ORGANIZATION AND FORMATTING:

Reviewer #1 (Comments to the Authors (Required)):

This study tested the impact of SHP2 inhibition in a model of PNF tumor progression. Using several approaches to measure tumor responsiveness and immune invasion, including flow cytometry and scRNA-seq, the inhibition of SHP2 protected mice from cancer to a similar degree as MEK inhibitors. Importantly, scRNA-seq from murine models had similar monocyte/macrophage response to new human scRNA-seq data generated by the authors. The mouse tumor size was associated with monocyte/macrophage numbers, suggesting a potential mechanism for the drug action in reducing macrophage numbers. This was also associated with enhanced T cell infiltration. Interestingly, the authors also showed that the time of day when the drugs are administered dramatically influenced the potency of the drug. Overall, this is an important study with several noteworthy findings. The study is performed rigorously and methods are very detailed. There were only a few concerns outlined below that would help to enhance the clarity of the paper.

1. Figure 1E imaging is too small and not at a high enough resolution to interpret the staining. Similarly, the flow cytometry plots in the supplement figure are difficult to evaluate due to low resolution. Please improve these so that it can be clearly visualized.
2. I couldn't find a reference to Fig. 4F-G in the main text. I believe it is mistakenly referenced as 4D in line 483.
3. The flow cytometry gating for Fig 5E is hard to determine what is being shown. Please show the full gating strategy for this experiment.
4. The flow cytometry gating strategy for tumor monocytes and macrophages appears to be somewhat superficial. There doesn't appear to be a definitive monocyte vs macrophage antibody gating presented that would allow for a clear delineation of these cells. The gating scheme shown in the supplement does not show how nonclassical monocytes, macrophages, and MDSCs are separated from one another. If they cannot be separated, please include this discussion point in the text so that a reader is not confused about the data being presented.
5. The nomenclature for the Ly6c+ CD62L+ monocyte is unclear why they are being called "immature/inflammatory", and why the Ly6c- monocytes are being called "reparative, long-lived". Did the cells show specific features that support this naming? It may make sense to use common nomenclature of classical and nonclassical monocytes for these subsets, or just reference as Ly6C+ and Ly6C-, without assigning functional attributes if they are not tested in this study.
6. Formatting for the Reference section is inconsistent with several errors. This includes reference #1 (line 995) "Barkal SHP2 therapy", and Line 1138: Lázaro. Beyond these, the formatting for several of the citations is presented in a different style than others. They should be uniform.
7. While most of the writing in the paper is very nice, the abstract should be revisited to improve readability/flow, and PNF should be defined.

Reviewer #2 (Comments to the Authors (Required)):

Ahmari et al, Immune modulation by time-of-1 day dosing determines the efficacy of SHP2 inhibition

In this manuscript, the authors test whether SHP2i results in decreased RAS activity that is associated with anti-tumor effect as well as alteration of the PNF tumor immune microenvironment. They also found that diurnal patterns of monocyte trafficking led to differential SHP2i effects driven by time of day of administration, supporting the notion that SHP2i targeting monocyte derived tumor macrophages leads to tumor shrinkage. They use analysis of RNAseq data and immune cell deconvolution to arrive at conclusions regarding the effects of SHP2i, compared to or in combination with MEKi, on the immune cells, as well as the combination with anti-PD1. The immune cell analysis is enlightening but mostly descriptive and in line with prior literature, although not necessarily of these drugs in this disease state. The figures and data generated are of reasonable quality and described in adequate detail, although there are a few areas that can be clarified further.

Major comments

As the primary effects of SHP2i in NF1-deficient RAS driven cancers is expected to be driven through RAS-ERK signaling, but

the authors see different effects when using SHP2i versus MEKi, can the authors comment or speculate on what is responsible for these differences? It would be nice to offer some speculation as Discussion.

In some figures, MEKi is GDC-0973, aka cobimetinib (Fig 1A) - in others, PD0325901 (aka mirdametinib), and in 1H - PD98059 (unusual), but in many figures it is simply MEKi. As the authors have used at least 2, maybe 3, different MEKi in the experiments, the figures, or at least the legends, need to state what is being tested clearly (ie, fig 4 legend, all just MEKi). Why call cobimetinib GDC-0973 in some experiments and cobimetinib in others?

Figure 1H - with all of the available clinically active MEKi (which one can buy on any commercial site), why did the authors choose PD98059, which is an old tool compound that pre-dated those that advanced to human trials and had micro-molar potency. The results would be more relevant with any of the currently available nanomolar potency allosteric compounds such as trametinib or mirdametinib. Actually - results, line 366 - the authors state PD0325901 when referencing figure 1H, so they need to clarify which one is accurate. The figure legend does not state which drug is used.

Minor comments:

The authors should proofread carefully, as several sentences have missing words or awkward structure, ie, line 62, others. I don't understand what is being shown with the white bars (empty) and black (filled) in figure 7B.

When citing the published literature on SHP2i in MPNST, the authors should also add the paper by Wang et al, describing SHP2i in combination with CDK4/6i, PMID 38000020.

The references are incompletely formatted (ie line 995, 1017, as examples, there may be others).

For better readability, the authors should revise their figure labels to use consistent font sizes, and check spelling (vehicle, Fig 3B, for example). As an example the boxed label in 4a is a different font size from 4B, and then 4F is quite a bit larger, all of the inconsistencies are distracting from the content; similar, Fig 5G and 5H.

Reviewer #3 (Comments to the Authors (Required)):

In the course of their characterization and assessment of the functional contribution of the immune microenvironment to neurofibroma development and maintenance, the group of Nancy Ratner is now reporting on the effect of SHP2 inhibitor in the context of neurofibroma. This is a relevant study given the results published in other tumor type and presented in the introduction section. On one side, there is a number of interesting findings reported in this manuscript that deserve to be published :

-SHP2 target validation in neurofibroma mouse model

-immune cell population change upon SHP2 inhibitor treatment

-circadian rhythm-dependent anti-tumor effect of SHP2 inhibitor

On the other side, the manuscript is difficult to follow in general. The divergence into different direction/story (PD1 and T cells, circadian rhythm dependency) that do not help the focus. The title only highlight a fraction of the data presented in this manuscript. The new title should contain the keyword PNF and better represent the findings of the authors. In addition, I think there is a too high emphasis on the comparison SHP2 vs MEK inhibition. Characterizing SHP2 as a new target in neurofibroma is something new by itself. Some efforts should be made to present the SHP2 finding first and in a second step, the results comparing it to MEK inhibition to highlight similarities and difference. This effort could even be done using the current figure set but by adding extra sentences in the main text to first present the results of SHP2 inhibition before jumping into the SHP2 vs MEK inhibition comparison. Also, there is a lack of proof-reading that is detail below that do not help the reader and do not help the author to make their story shine.

More specifically :

1) The authors used two complementary techniques to characterize immune cells. scRNAseq as presented in Fig.2 and FACS as presented in Fig. 3. One of the conclusion of Fig. 2 is that immune cell population is drastically decreased upon SHP2 inhibition. One of the conclusion of Fig 3 is that some population are expanded (macrophages) and some are shrunked (T cells). I think Fig 4 is attempting to use Fig 3 findings highlight within the Fig 2 scRNAseq dataset but I am not sure.

2) CD163- and CD163+ interpretation. Is CD163- and CD163+ refer to M1 and M2 macrophages? Why the nomenclature M1/M2 for macrophages is not used? In Fig. 4E CD163- are up in PNF compare to WT nerve but it seem to contradict data from Fig. S1DE where CD163- appear down in PNF compare to WT nerve. In Fig 4E if considering only WT. Why the sum of CD163- and CD163+ do not add up to 100%? (CD163- represent 5% and CD163+ 15%). What is the remaining 80%?

3) Waterfall plot present data from individual tumors. How tumor volume was follow for individual tumors?

4) The sentence ``Conversely, giving the drug at night or combining it with the immune checkpoint inhibitor anti-PD1 reverses or negates SHP2 inhibitor mediated tumor shrinkage by changing monocytes and macrophage phenotypes rather than T-cell frequency`` appear rather correlative to me. What is the best evidence directly addressing the cause to effect of changing monocytes and macrophage reverse or negates SHP2 inhibitor mediated shrinkage to conclude this way?

5) References. In some occasion, the reference cited within the main text do not match the one found at the end in the reference

section. The references at the end do not follow the same presentation style. Please take this opportunity to fully revised the references. Here is some examples :

- Ref #5. page 5 Brosseau et al., at the end Brosseau et al.
- Ref #6. Page38 Cristofides et al., page 29 Christophides et al.
- Ref #29 Page 19 Lund et al. 2023, at the end Lund et al. 2024
- Ref #1, 9 appear to be incomplete

6) Poor homogeneity in nomenclature/abbreviation. Please take this opportunity to fully revised the homogeneity throughout the text and figures. Here is some examples :

- When referring to drug used. Its only in the discussion section when I realized that more than one MEK inhibitor was used in this study : ``We used MEK inhibitor PD0325901 for most experiments...
- line 406 MEK1, line 411 MEKi
- The cell-type specific markers used are not sufficiently presented/justified throughout the paper to allow a smooth reading. Sometimes the cell type is display without explicit used of specific markers, sometimes markers are presented without referring to a particular cell type subset. For example when referring to macrophages Page 33, line 773. Mf?. What is the marker used?

8) Comments on figures and their legends

- Page 32 Fig 2 legend. The text from line 752 to 762 do not match the figure 2 provided in the figure set.
- Page 19, line 441. Text is pointing to figure 3E but there is no such figure panel in the figure set provided
- Suppl Fig 2 legend is entirely missing
- Page 20 line 456. The r value do not match the one shown in the figure 3D (0.5479)
- Fig 1E. Please provide larger/higher magnification IHC panels
- Fig. 3 typo in ``vehicle``
- Fig. S4 what is normal, what is PNF?
- Fig. 5AB. What is durability?
- Fig. 6DE. Why WT nerve is not shown as in panel C?
- Fig. 7E. Is the circadien dependent known for this immune cell type?
- Fig. S1DE. What is panel E. Some genes are redundant from panel D
- Fig. S1D. Use the same nomeclature everywhere (PNF, not PN)
- Fig. S3. Consider spreading A, B and C on 3 separate suppl figures as the panel are unreadable
- Fig. S7AB. Please add the color code legend as in other suppl figures.
- Fig S7D. Isnt it activated Tcells?

Other :

- Page 6. Line 122-124. Please rephrase.
- Page 32 line 751. Should be ``immune`` cells, not ``Schwann`` cell.

Overall, although I have expertise and interest into this manuscript, the overall data presentation prevent me to go into more details

To the editor:

We thank the reviewers for their constructive comments, and their support of our manuscript. We have made all the changes suggested by the reviewers per the point-by-point response below, and hope that the manuscript will now be acceptable for publication.

Reviewer #1 (Comments to the Authors (Required)):

This study tested the impact of SHP2 inhibition in a model of PNF tumor progression. Using several approaches to measure tumor responsiveness and immune invasion, including flow cytometry and scRNA-seq, the inhibition of SHP2 protected mice from cancer to a similar degree as MEK inhibitors. Importantly, scRNA-seq from murine models had similar monocyte/macrophage response to new human scRNA-seq data generated by the authors. The mouse tumor size was associated with monocyte/macrophage numbers, suggesting a potential mechanism for the drug action in reducing macrophage numbers. This was also associated with enhanced T cell infiltration. Interestingly, the authors also showed that the time of day when the drugs are administered dramatically influenced the potency of the drug. Overall, this is an important study with several noteworthy findings. The study is performed rigorously and methods are very detailed. There were only a few concerns outlined below that would help to enhance the clarity of the paper.

1. Figure 1E imaging is too small and not at a high enough resolution to interpret the staining. Similarly, the flow cytometry plots in the supplement figure are difficult to evaluate due to low resolution. Please improve these so that it can be clearly visualized.

Response: We thank the reviewer for pointing this out; we have improved the resolution of both figures.

2. I couldn't find a reference to Fig. 4F-G in the main text. I believe it is mistakenly referenced as 4D in line 483.

Response: We made the correction.

3. The flow cytometry gating for Fig 5E is hard to determine what is being shown. Please show the full gating strategy for this experiment.

Response: The full gating is now shown in supplemental figure 3A.

4. The flow cytometry gating strategy for tumor monocytes and macrophages appears to be somewhat superficial. There doesn't appear to be a definitive monocyte vs macrophage antibody gating presented that would allow for a clear delineation of these cells. The gating scheme shown in the supplement does not show how nonclassical monocytes, macrophages, and MDSCs are separated from one another. If they cannot be separated, please include this discussion point in the text so that a reader is not confused about the data being presented.

Response: We thank the reviewer for raising this important concern. As shown in Supplemental Figure 4G, we used FlowSOM clustering to resolve distinct tumor-infiltrating myeloid populations, including classical monocytes, macrophages, and myeloid-derived suppressor cells (MDSCs). Clusters were annotated based on canonical marker expression: Ly6C⁺Ly6G⁺ for MDSCs, Ly6C⁺Ly6G⁻ for monocytes, and Ly6C⁻Ly6G⁻F4/80⁺CD11b⁺CD64⁺CD14⁺CD172a⁺ for tumor-associated macrophages. Classical monocytes were further distinguished by high Ly6C expression. These color-coded clusters were then back-gated onto two-dimensional flow plots, demonstrating that conventional gating could reliably recapitulate the populations identified through unsupervised clustering. The reviewer is correct that nonclassical monocytes (Ly6C^{lo}) are not clearly separable from macrophages based on surface markers alone in the tumor context, and this limitation is now noted in the main text. Nonetheless, the alignment between high-dimensional clustering and traditional gating confirms the robustness of our classification strategy and supports the interpretation of monocyte and macrophage subsets in our dataset. We have clarified this methodological approach in the revised manuscript for transparency.

5. The nomenclature for the Ly6c⁺ CD62L⁺ monocyte is unclear why they are being called "immature/inflammatory", and why the Ly6c⁻ monocytes are being called "reparative, long-lived". Did the cells show specific features that support this naming? It may make sense to use common nomenclature of classical and nonclassical monocytes for these subsets, or just reference as Ly6C⁺ and Ly6C⁻, without assigning functional attributes if they are not tested in this study.

Response: We thank the reviewer for bringing this up; we altered the writing to reference them as Ly6C⁺ and Ly6C⁻; in one figure legend they are further annotated as classical and nonclassical monocytes.

6. Formatting for the Reference section is inconsistent with several errors. This includes reference #1 (line 995) "Barkal SHP2 therapy", and Line 1138: Lázaro. Beyond these, the formatting for several of the citations is presented in a different style than others. They should be uniform.

Response: We thank the reviewer for pointing this out. This has been corrected.

7. While most of the writing in the paper is very nice, the abstract should be revisited to improve readability/flow, and PNF should be defined.

Response: We thank the reviewer for the comment. The abstract has been thoroughly revised.

Reviewer #2 (Comments to the Authors (Required)):

Ahmari et al, Immune modulation by time-of-1 day dosing determines the efficacy of SHP2 inhibition

In this manuscript, the authors test whether SHP2i results in decreased RAS activity that is associated with anti-tumor effect as well as alteration of the PNF tumor immune microenvironment. They also found that diurnal patterns of monocyte trafficking led to differential SHP2i effects driven by time of day of administration, supporting the notion that SHP2i targeting monocyte derived tumor macrophages leads to tumor shrinkage. They use analysis of RNAseq data and immune cell deconvolution to arrive at conclusions regarding the effects of SHP2i, compared to or in combination with MEKi, on the immune cells, as well as the combination with anti-PD1. The immune cell analysis is enlightening but mostly descriptive and in line with prior literature, although not necessarily of these drugs in this disease state. The figures and data generated are of reasonable quality and described in adequate detail, although there are a few areas that can be clarified further.

Major comments

1. As the primary effects of SHP2i in NF1-deficient RAS driven cancers is expected to be driven through RAS-ERK signaling, but the authors see different effects when using SHP2i versus MEKi, can the authors comment or speculate on what is responsible for these differences? It would be nice to offer some speculation as Discussion.

Response: We now state in the discussion... "Yet other effects of SHP2 inhibition differed from those of MEK inhibition. SHP2 is a phosphatase with many substrates within and outside the RAS-MAPK pathway, and de-phosphorylation of one or more such substrates likely accounts for differences in effects between MEK and SHP2 inhibition. For example, SHP2 inhibition, but not MEK inhibition, altered Nur77 expression on monocytes from tumor-bearing mice and changed aspects of MDSC transcriptional profiles..."

2. In some figures, MEKi is GDC-0973, aka cobimetinib (Fig 1A) - in others, PD0325901 (aka mirdametinib), and in 1H - PD98059 (unusual), but in many figures it is simply MEKi. As the authors have used at least 2, maybe 3, different MEKi in the experiments, the figures, or at least the legends, need to state what is being tested clearly (ie, fig 4 legend, all just MEKi).

Response: We thank the reviewer for bringing this to our attention. We have now specified the MEK inhibitor used in each case, either within the figures or their legends. The previous mention of PD98059 was an error (it was PD0325901) which has been corrected.

3. Why call cobimetinib GDC-0973 in some experiments and cobimetinib in others?

Response: GDC-0973 is the compound name for cobimetinib; both refer to the same drug. We now introduce it as GDC-0973/cobimetinib and subsequently use the common name.

4. Figure 1H - with all of the available clinically active MEKi (which one can buy on any commercial site), why did the authors choose PD98059, which is an old tool compound that pre-dated those that advanced to human trials and had micro-molar potency. The results would be more relevant with any of the currently available nanomolar potency allosteric compounds

such as trametinib or mirdametinib. Actually - results, line 366 - the authors state PD0325901 when referencing figure 1H, so they need to clarify which one is accurate. The figure legend does not state which drug is used.

Response: We appreciate the reviewer's careful reading. PD0325901 was the compound used in Figure 1H. The mention of PD98059 was an error in the figure legend, now corrected. PD98059 was not used in any experiments.

Minor comments:

5. The authors should proofread carefully, as several sentences have missing words or awkward structure, ie, line 62, others.

Response: We revised around line 62, which now reads "Biallelic inactivation of the tumor suppressor gene *Neurofibromatosis type 1 (NF1)* in nerve Schwann cells results in formation of benign peripheral nerve tumors known as neurofibromas (Serra et al., 2000; Pemov et al., 2017). Plexiform neurofibromas (PNF) form in deep nerves and are found in at least half of individuals with NF1."

6. I don't understand what is being shown with the white bars (empty) and black (filled) in figure 7B. Response: This is showing when the mice were treated. Mice were treated with SHP2i or vehicle at 9am or 9pm. If the bar is black, they were treated at night. If the bar is white they were treated in the morning.

The legend for 7B now reads: "Black bars indicate animals treated at night; while bars indicate animals treated in the morning."

7. When citing the published literature on SHP2i in MPNST, the authors should also add the paper by Wang et al, describing SHP2i in combination with CDK4/6i, PMID

38000020. Response: The study by Wang et al. (PMID: 38000020), which investigates SHP2 inhibition in combination with CDK4/6 inhibition in MPNST, has now been added to the manuscript as an additional reference to strengthen the context of SHP2-targeted therapies.

8. The references are incompletely formatted (ie line 995, 1017, as examples, there may be others). Response: References have been re-formatted.

9. For better readability, the authors should revise their figure labels to use consistent font sizes, and check spelling (vehicle, Fig 3B, for example). As an example the boxed label in 4a is a different font size from 4B, and then 4F is quite a bit larger, all of the inconsistencies are distracting from the content; similar, Fig 5G and 5H.

Response: We thank the reviewer for this helpful feedback. All figure labels have been revised for consistency, and are now presented in Arial font, size 18, to improve readability. We have also corrected the spelling error in Figure 3B ("vehicle") and carefully reviewed all figures to ensure consistent formatting across panels, including Figures 4A, 4B, 4F, 5G, and 5H.

Reviewer #3 (Comments to the Authors (Required)):

In the course of their characterization and assessment of the functional contribution of the immune microenvironment to neurofibroma development and maintenance, the group of Nancy Ratner is now reporting on the effect of SHP2 inhibitor in the context of neurofibroma. This a relevant study given the results published in other tumor type and presented in the introduction section. On one side, there is a number of interesting findings reported in this manuscript that deserve to be published :

- SHP2 target validation in neurofibroma mouse model
- immune cell population change upon SHP2 inhibitor treatment
- circadian rhthym-dependent anti-tumor effect of SHP2 inhibitor

On the other side, the manuscript is difficult to follow in general. The divergence into different direction/story (PD1 and T cells, circadian rhthym dependency) that do not help the focus. The title only highlight a fraction of the data presented in this manuscript.

1. **The new title should contain the keyword PNF nd better represent the findings of the authors.**

We revised the title to read: “Characterization of neurofibroma immune cells reveals that immune modulation by time-of-day dosing determines the efficacy of SHP2 inhibition” to better reflect the major findings of the study. The term PNF is, we believe, too much field jargon to add to the title.

2. I think there is a too high emphasis on the comparison SHP2 vs MEK inhibition. Characterizing SHP2 as a new target in neurofibroma is something new by itself. Some efforts should be made to present the SHP2 finding first and in a second step, the results comparing it to MEK inhibition to highlight similarities and difference. This effort could even be done using the current figure set but by adding extra sentences in the main text to first present the results of SHP2 inhibition before jumping into the SHP2 vs MEK inhibition comparison.

Response: We gave significant consideration of this presentation option prior to manuscript submission. Upon consideration, and consulting many clinicians, we felt it important to show a direct comparison of the agents. Therefore, we maintained the figure set as in the 1st submission. However, to address this reviewer’s concern, we re-arranged the description of Figure 1 to read “We treated DhhCre;Nf1fl/fl mice with a SHP2 inhibitor (10 mg/kg or 30 mg/kg RMC-4550; q.d., p.o., SHP2i), a MEK inhibitor (5mg/kg GDC-0973/cobimetinib; q.d., p.o., MEKi), or the combination, each on a 5-days-on, 2-days-off schedule. Tumors in vehicle-treated mice grew over the 60-day treatment period (**Figure 1A**).

3. Also, there is a lack of proof-reading that is detail below that do not help the reader and do not help the author to make there story shine.

More specifically :

- 1) The authors used two complementary techniques to characterize immune cells.

scRNAseq as presented in Fig.2 and FACS as presented in Fig. 3. One of the conclusion of Fig. 2 is that immune cell population is drastically decreased upon SHP2 inhibition. One of the conclusion of Fig 3 is that some population are expanded (macrophages) and some are shrunked (T cells). I think Fig 4 is attempting to use Fig 3 findings highlight within the Fig 2 scRNAseq dataset but I am not sure.

Response: We thank the reviewer for their careful evaluation. We apologize for the confusion caused by the presentation of Figures 2 and 3. To clarify, both the scRNA-seq (Figure 2) and flow cytometry (Figure 3) analyses consistently demonstrate a reduction in immune cell populations following SHP2 inhibition. Specifically, in Figure 2B, scRNA-seq data show a marked decrease in the percentage of immune cells relative to total live cells in SHP2i-treated tumors. Similarly, Figure 3B confirms this reduction across multiple immune populations by flow cytometry, with macrophages, dendritic cells, and monocytes all significantly decreased. Notably, T cells increase slightly as a percentage of remaining CD45+ cells but not in absolute number. Figure 4 builds on the flow cytometry data (Figure 3) to further dissect macrophage subtypes and validate their reduction using dimensionality reduction and clustering, rather than integrating with the scRNA-seq data from Figure 2. We have revised the text to more clearly delineate how each figure builds on distinct methodologies and findings.

2) CD163- and CD163+ interpretation. Is CD163- and CD CD163+ refer to M1 and M2 macrophages? Why the nomenclature M1/M2 for macrophages is not used?

Response: We appreciate the reviewer's close reading. The CD163⁺ and CD163⁻ populations in our analysis do not correspond to classical M1/M2 polarization states, which we intentionally avoid given the oversimplification and limited relevance of that nomenclature in tissue macrophages and tumor settings (Murray et al., 2014; Lavin et al., 2017). Instead, we use CD163 expression to distinguish between resident (CD163⁺) and monocyte-derived (CD163⁻) macrophages, consistent with prior work in peripheral nerve and neurofibroma (Lund et al., 2024).

The text now reads: "CD163 expression distinguishes resident (CD163⁺) and monocyte-derived (CD163⁻) macrophages in peripheral nerve and neurofibroma (Lund et al., 2024). Of note, the nerve CD163⁺ and CD163⁻ populations do not correspond to M1/M2 polarization states, which are of limited relevance to tissue macrophages and tumor settings (Murray et al., 2014; Lavin et al., 2017)."

Regarding Figure 4E and Suppl. Fig. 1D-E, these plots do not contradict one another. Figure 4E shows flow cytometry data analyzed as the percentage of each macrophage subset out of total CD45⁺ immune cells. In contrast, Suppl. Fig. 1D-E presents gene expression data from scRNA-seq, analyzed at the level of individual cell

clusters, focusing on transcript abundance (e.g., Cd163 mRNA levels). We clarified this in the text/figure legends to avoid confusion.

The title for Figure 4 now reads: **Figure 4: Flow cytometric analysis shows that SHP2 inhibition alters activation of tumor macrophages which are predominately CD163 negative.** Supplemental Figure 1 legend now reads: We then compared wild type and tumor macrophage clusters (**Supp. Fig. 1C**), and tumor macrophages (vehicle versus drug treatment), focusing on genes encoding markers used to discriminate macrophage types in flow cytometry (**Supp. Figs. 1D, E**). These markers were similarly analyzed in tumor macrophages; *Cd163* expression was reduced in tumour macrophages and showed relative increases after drug treatment (**Supp. Fig. 1C, D**). Changes in transcription of markers of activation were also caused by drug treatment (**Supp. Figure 1E**). This information is also summarized in the text.

In Fig. 4E CD163⁻ are up in PNF compare to WT nerve but it seem to contradict data from Fig. S1DE where CD163⁻ appear down in PNF compare to WT nerve. In Fig 4E if considering only WT. Why the sum of CD163⁻ and CD163⁺ do not add up to 100%? (CD163⁻ represent 5% and CD163⁺ 15%). What is the remaining 80%?

In Fig 4E if considering only WT. Why the sum of CD163⁻ and CD163⁺ do not add up to 100%? (CD163⁻ represent 5% and CD163⁺ 15%). What is the remaining 80%?

Regarding this apparent discrepancy in proportions, we present immune cell subsets (including CD163⁺ and CD163⁻ macrophages) as a percentage of either total live cells or total CD45⁺ cells, not as a percentage of macrophages. Therefore, the sum of CD163⁺ and CD163⁻ macrophages will not add up to 100%- the remaining cells consists of other immune /non-immune cells. This is now clarified in the legend to Suppl. 4E.

3) Waterfall plot present data from individual tumors. How tumor volume was follow for individual tumors?

Response: We thank the reviewer for this question. Tumor volume for individual tumors was measured using volumetric MRI, a method well established in our lab and previously described in detail (Wu et al., 2012; Jessen et al., 2013). Briefly, tumors were imaged in anesthetized mice using a 7T Bruker Biospec MRI system. Serial images were acquired in three planes to accurately define tumor boundaries, and volumes were calculated from outlined tumor areas and slice thickness. This approach allows longitudinal tracking of tumor volume in the same animal over time.

In the methods section we now state: "*Magnetic Resonance Imaging and analysis was performed in anaesthetized mice as described, using a 7T Bruker Biospec System to acquire images in 3 planes to position the 3D volume (Wu et al., 2012). Tumour volumes were measured in mice at 5, 7, and 9 months old, with treatment between 7 and 9 months of age, or*

off drug as noted in the text. We calculated tumor volume from the area of graphic outlines and MRI slice thickness.”

We have added a methodological clarification to the figure legend. It now states for 1A: “Waterfall plot illustrating percent change in tumor volume during the 60-day dosing period based on consecutive MRI scans, demonstrating similar tumor shrinkage in treatment groups. Each bar represents the percentage change in tumor volume for an individual mouse.”

4) The sentence “Conversely, giving the drug at night or combining it with the immune checkpoint inhibitor anti-PD1 reverses or negates SHP2 inhibitor mediated tumor shrinkage by changing monocytes and macrophage phenotypes rather than T-cell frequency” appear rather correlative to me. What is the best evidence directly addressing the cause to effect of changing monocytes and macrophage reverse or negates SHP2 inhibitor mediated shrinkage to conclude this way?

Response:

We agree that this is only a correlation. We revised the sentence to read: “Conversely, giving the drug at night or combining it with the immune checkpoint inhibitor anti-PD1 reverses or negates SHP2-inhibitor-mediated tumor shrinkage, correlating with changing monocyte and macrophage phenotypes rather than T-cell frequency. While more evidence is necessary to thest addressing whether changing monocytes and/or macrophage reverse or negates SHP2 inhibitor mediated shrinkage, these results suggest that SHP2-regulated monocyte activation and circadian influences on immune cell trafficking critically shape neurofibroma growth and treatment response.”

5) References. In some occasion, the reference cited within the main text do not match the one found at the end in the reference section. The references at the end do not follow the same presentation style. Please take this opportunity to fully revised the references. Here is some examples :

-Ref #5. page 5 Brousseau et al., at the end Brosseau et al.

-Ref #6. Page38 Cristofides et al., page 29 Christophides et al.

-Ref #29 Page 19 Lund et al. 2023, at the end Lund et al. 2024

-Ref #1, 9 appear to be incomplete

Response: We carefully reviewed and revised the reference list to ensure that the in-text citations match the corresponding entries in the reference section. This includes correcting author name spelling (e.g., Brousseau vs. Brosseau; Cristofides vs. Christophides), publication year (e.g., Lund et al. 2023 vs. 2024), and completing the incomplete references (e.g., Refs #1 and #9). We also standardized formatting throughout the reference section to ensure consistency in presentation.

6) Poor homogeneity in nomenclature/abbreviation. Please take this opportunity to fully revised the homogeneity throughout the text and figures. Here is some examples :

-When referring to drug used. Its only in the discussion section when I realized that more than one MEK inhibitor was used in this study : ``We used MEK inhibitor PD0325901 for most experiments...

Response: We thank the reviewer for this comment. We have reviewed the manuscript to ensure consistent use of nomenclature and abbreviations throughout. The specific MEK inhibitor used in each experiment is now clearly indicated in the corresponding figure or figure legend.

-line 406 MEKI, line 411 MEKi

Response: We thank the reviewer for pointing this out. This has been corrected for consistency; all instances now use "MEKi."

-The cell-type specific markers used are not sufficiently presented/justified throughout the paper to allow a smooth reading. Sometimes the cell type is display without explicit used of specific markers, sometimes markers are presented without referring to a particular cell type subset. For example when referring to macrophages Page 33, line 773. Mf?. What is the marker used?

Response: To improve clarity, we updated the main text to specify that macrophages are defined as shown in Supplemental Figures 3–4--See lines 450-480.

Specifically, in our flow cytometry and clustering analyses, macrophages were identified by sequential gating on CD11b⁺ cells with low side scatter, exclusion of Ly6C⁺ Ly6G⁺ populations (monocytes and MDSCs), and positive expression of CD14 and CD172a to exclude most dendritic cells. Final gating on F4/80^{hi} and stratification by CD163 expression allowed resolution of two major macrophage subsets, as shown in Supplemental Figure 4G. We reviewed the manuscript to ensure that immune cell populations are consistently associated with either their marker definitions or referenced figure panels.

8) Comments on figures and their legends

Page 32 Fig 2 legend. The text from line 752 to 762 do not match the figure 2 provided in the figure set.

Response: We thank the reviewer for bringing this to our attention. The legend for Figure 2 has been revised to accurately reflect the current version of the figure. The discrepancy between the text and figure has now been corrected.

Page 19, line 441. Text is pointing to figure 3E but there is no such figure panel in the figure set provided

Response: We thank the reviewer for catching this error. The text was intended to refer to Supplemental Figure 4E, not Figure 3E. This has been corrected in the revised

manuscript.

Suppl Fig 2 legend is entirely missing

Response: We thank the reviewer for noting this omission. The legend for Supplemental Figure 2 has now been added to the revised manuscript.

Page 20 line 456. The r value do not match the one shown in the figure 3D (0.5479)

Response: We thank the reviewer for pointing this out. The r value in the text has been corrected to match the value shown in Figure 3D (0.5479).

Fig 1E. Please provide larger/higher magnification IHC panels

Response: We thank the reviewer for this suggestion. Higher magnification IHC panels for Figure 1E have been provided in the revised figure to improve clarity and visualization of tissue architecture.

Fig. 3 typo in ``vehicle``

Response: We thank the reviewer for pointing out this typo. It has been corrected in the revised version of Figure 3.

Fig. S4 what is normal, what is PNF?

Response: We thank the reviewer for this comment. Labels have been added to Supplemental Figure 4B to clearly indicate that the left panel represents PNF and the right panel represents normal (WT) nerve.

Fig. 5AB. What is durability?

Response: We thank the reviewer for this question. In Figure 5A–B, “durability” refers to tumors treated with the SHP2 inhibitor RMC-4550 for 30 days followed by a 30-day treatment-free period. This group was included to evaluate whether the immune changes observed during treatment persist after withdrawal. We have clarified this definition in the figure legend.

Fig. 6DE. Why WT nerve is not shown as in panel C?

Response: We thank the reviewer for the question. As shown in Figure 5F, we include WT nerve comparisons for both Ly6C^{hi} and Ly6C^{lo} monocytes. However, for Panel D, WT nerve data are not available for this specific analysis. We have clarified this in the figure legend to avoid confusion.

Fig. 7E. Is the circadian dependent known for this immune cell type?

Response: Circadian regulation of Ly6C^{hi} monocytes has been reported (e.g., Nguyen et al., *Immunity*, 2013). Less is known about circadian patterns in Ly6C^{lo} CD62L⁻ monocytes. Our data suggest that both subsets exhibit circadian fluctuations in the tumor context, highlighting a potential tumor-driven alteration of normal rhythmicity. The text now reads: In wild type mice, Ly6C^{low};CD62L⁻ monocytes were reduced in circulation at night, consistent with prior reports of circadian regulation of monocyte subsets (Nguyen et al., 2013); this diurnal pattern was lost in tumor-bearing mice.

Fig. S1DE. What is panel E. Some genes are redundant from panel D

Response: The only redundant gene is Cd163. See response above.

Fig. S1D. Use the same nomenclature everywhere (PNF, not PN)

Response: Done.

Fig. S3. Consider spreading A, B and C on 3 separate suppl figures as the panel are unreadable

Response: We increased the figure 3 resolution and adjusted the layout of panels A–C in Figure S3 to improve readability. We believe the revised figure is now clear while maintaining the grouping of related data.

Fig. S7AB. Please add the color code legend as in other suppl figures.

Response: We thank the reviewer for noting this. The color code legend has been added to Figure S7A–B for consistency with the other supplementary figures.

Fig S7D. Isnt it activated Tcells?

Response: We appreciate the reviewer's comment. Figure S7D shows subsets of monocytes, specifically Ly6C^{hi} CD62L⁻ and Ly6C^{low} CD62L⁺ populations. T cells were not analyzed in this panel. We have clarified this in the figure legend to avoid confusion.

Other :

Page 6. Line 122-124. Please rephrase.

Response: The last line of the introduction now reads "Our results demonstrate the therapeutic potential of SHP2 inhibition as a single agent in neurofibromas and the immunomodulatory potential of these MEK and SHP2 inhibitors."

Page 32 line 751. Should be ``immune`` cells, not ``Schwann`` cell.

Response: Our error! Now corrected.

Overall, although I have expertise and interest into this manuscript, the overall data presentation prevent me to go into more details.

We hope that these revisions have provided sufficient changes in the presentation to enable this reviewer to better understand the manuscript.

July 30, 2025

RE: Life Science Alliance Manuscript #LSA-2025-03359-TR

Dr. Nancy Ratner
Cincinnati Children's Hospital Medical Center
Department of Pediatrics
3333 Burnet Avenue
Cincinnati, Ohio 45229-0713

Dear Dr. Ratner,

Thank you for submitting your revised manuscript entitled "Daytime SHP2 inhibitor dosing, when immune cell numbers are elevated, shrinks neurofibromas". As you will see, reviewers are satisfied with the changes in place. We would be happy to publish your paper in Life Science Alliance pending final revisions necessary to meet our formatting guidelines.

- Please add a Running Title in our system.
- Please add a Category for your manuscript in our system.
- Please add Keywords for your manuscript in our system.
- Please add a Summary Blurb/Alternate Abstract in our system.
- Please add the X and Bluesky handles of your host institute/organization as well as your own or/and one of the authors in our system.
- The title in the manuscript and on the submission page need to match; please change the title on the submission page to the current title in the manuscript file.
- Please consult our manuscript preparation guidelines <https://www.life-science-alliance.org/manuscript-prep> and make sure your manuscript sections are in the correct order and add a Data Availability section and a Conflict of Interest statement.
- Please use the [10 author names, et al.] format in your references (i.e. limit the author names to the first 10).
- Tables are mentioned in the manuscript text while no tables have been provided. Please rectify this discrepancy. Please upload your Tables in editable .doc or excel format. Main tables can be included at the end of manuscript file.
- Please add label F in the caption for Figure 7.
- Please add a callout for Figures: 7G-H, S3A-C, S4A-D,H, S5A-C,G,J, S7D,F X to your main manuscript text. There are callouts for Figure S3FG and S2C which are not present in the figures. Please rectify this discrepancy.
- Please add molecular weight markers to the blot in Figure 1D.
- The current abstract includes a potential discrepancy that may confuse readers. -The sentence on line 28, indicating tumor shrinkage by SHP2 inhibition, is somewhat inconsistent with the later statement on line 33, that tumors shrink only during daytime treatment. Please consider how to modify either or both statements for greater clarity.

LSA now encourages authors to provide a 30-60 second video where the study is briefly explained. We will use these videos on social media to promote the published paper and the presenting author (for examples, see <https://docs.google.com/document/d/1-UWCfbE4pGcDdcgzcmiuJl2XMBJnxKYeqRvLLrLSo8s/edit?usp=sharing>). Corresponding or first-authors are welcome to submit the video. Please submit only one video per manuscript. The video can be emailed to contact@life-science-alliance.org

A. FINAL FILES:

B. MANUSCRIPT ORGANIZATION AND FORMATTING:

Sincerely,

Reviewer #1 (Comments to the Authors (Required)):

All of my concerns have been addressed.

Reviewer #3 (Comments to the Authors (Required)):

My comments were answered satisfactorily.

2 more minors comments on figures

-Fig 1E. Please specify scale bar for the IHC images provided

-Fig 2E. On the MEKi T cells cluster 20. There is the letters ``Lor`` introduced at the bottom at the level of the word apoptosis. Not sure if it is meaningful.

September 8, 2025

RE: Life Science Alliance Manuscript #LSA-2025-03359-TRR

Prof. Nancy Ratner
Cincinnati Children's Hospital Medical Center
Department of Pediatrics, University of Cincinnati College of Medicine
3333 Burnet Avenue
#7013
Cincinnati, Ohio 45229-0713

Dear Dr. Ratner,

Thank you for submitting your Research Article entitled "Daytime SHP2 inhibitor dosing, when immune cell numbers are elevated, shrinks neurofibromas". It is a pleasure to let you know that your manuscript is now accepted for publication in Life Science Alliance. Congratulations on this interesting work and thank you once again for providing the additional western blot data we requested.

DISTRIBUTION OF MATERIALS:

Again, congratulations on a very nice paper. I hope you found the review process to be constructive and are pleased with how the manuscript was handled editorially. We look forward to future exciting submissions from your lab.

Sincerely,
